# Technological Potential Analysis and Vacant Technology Forecasting in Properties and Composition of Low-Sulfur Marine Fuel Oil (VLSFO and ULSFO) Bunkered in Key World Ports

Mikhail A. Ershov [1,*], Vsevolod D. Savelenko [1], Alisa E. Makhmudova [1], Ekaterina S. Rekhletskaya [2], Ulyana A. Makhova [1], Vladimir M. Kapustin [1,3], Daria Y. Mukhina [1,2] and Tamer M. M. Abdellatief [4]

[1] Department of Oil Refining Technology, Faculty of Chemical and Environmental Engineering, Gubkin Russian State University of Oil and Gas (National Research University), 119991 Moscow, Russia
[2] New Technologies Watch Center (NTWC LLC), 117546 Moscow, Russia
[3] Academy of Engineering, Peoples' Friendship University of Russia (RUDN University), 115419 Moscow, Russia
[4] Chemical Engineering Department, Faculty of Engineering, Minia University, EL-Minia 61519, Egypt
[*] Correspondence: m_ershov@fuelsdigest.com

**Abstract:** Analysis of the very-low-sulfur fuel oil (VLSFO) and ultra-low-sulfur fuel oil (ULSFO) bunkered in key ports in Asia, the Middle East, North America, Western Europe, and Russia is presented. The characteristics of said fuels, including density, sulfur content, kinematic viscosity, aluminum and silicon content, vanadium and nickel content, as well as pour point are investigated. Furthermore, the main trends and correlations are also discussed. Based on the graphical and mathematical analysis of the properties, the composition of the fuels is predicted. The key fuel components in Asian ports, the most important of which is Singapore, are hydrodesulfurized atmospheric residues (AR) (50–70%) and catalytic cracker heavy cycle oil (HCO) (15–35%) with the addition of other components, which is explained by the presence of a number of large oil refining centers in the area. In the Middle East ports, the most used VLSFO compositions are based on available resources of low-sulfur components, namely hydrodesulfurized AR, the production facilities of which were recently built in the region. In European ports, due to the relatively low sulfur content in processed oils, straight-run AR is widely used as a component of low-sulfur marine fuels. In addition, fuels in Western European ports contain on average significantly more hydrotreated vacuum gas oil (21%) than in the rest of the world (4–5%). Finally, a mixture of hydrotreated (80–90%) and straight-run fuel oil (10–15%) with a sulfur content of no more than 2.0–2.5% is used as the base low-sulfur component of marine fuels in the ports of Singapore and the Middle East.

**Keywords:** marine fuel; very-low-sulfur fuel oil (VLSFO); ultra-low-sulfur fuel oil (ULSFO); world ports; marine energy; fuel oil

## 1. Introduction

Fossil petroleum fuel development produces big quantities of harmful greenhouse gas exhaust emissions, which are connected to the global warming issue [1–5]. Their consumption and generation lead to significant quantities of harmful greenhouse gas exhaust emissions [6–10]. When talking about the production and consumption of marine (bunker) fuel, two kinds of fuels are generally looked upon—residual and distillate fuels [11–15]. In addition, the continuous growth of sea transport leads to a rise in climate pollution of sea regions annually [16]. In 2019, eleven billion tons of transport were transferred by sea, that needs about two hundred thirty-three million tons of marine fuel annually. Additionally, three hundred million tons of oil are turned into marine fuel per year in the world according to the report from the International Maritime Organization (IMO) [17]. IMO is accountable

for the enhancement of global shipping for ecological protection and safety. The most recent regulations involve the control of fuel standards and the decrease in harmful emissions of nitrogen oxides, carbon oxides, and sulfur oxides. Present limits, introduced in 2020, require decreasing the amount of sulfur in fuel to 0.5 by weight percent in the open sea and to 0.1 by weight percent in Special Environmental Control Areas (SECAs). Recently, shipping manufacturing has had various options for the utilization of low-sulfur residue marine fuels corresponding with ecological demands, involving the utilization of fuel surrogates [18,19].

One way to produce low-sulfur residue marine fuels is selective compounding, including blending distilling low-sulfur streams and residual sulfur. The key issue is acquiring the sedimentation stability of the compounded residual marine fuel [20]. Optimizing the composition of marine fuel is complex due to the amount of sulfur and ecological demands. Consequently, it forces manufacturers to utilize incompatible streams in the blend [21].

Residual fuels are obtained by blending heavy petroleum products (atmospheric residue (AR), vacuum residue (VR), heavy gas oils from thermal and thermo-catalytical processes) and diesel fractions of both straight-run and secondary conversion origin [22–27]. This type of fuel is utilized in marine boiler plants, as a motor fuel for low- and medium-speed diesel engines and fuel for gas turbine plants [28–30]. Distillate marine fuel is made from diesel fractions with the addition of secondary process light gas oils. This type of fuel is intended for use in marine high- and medium-speed diesel engines, as well as in gas turbine plants [31–33].

The main regulatory document for marine fuels in the world is the international standard ISO 8217:2017 "Petroleum products-Fuel (class F)-Specification of marine fuels" [34,35]. The current standard has replaced the similar one of ISO 8217:2012 [36] with changes related only to distillate fuels, including new grades have been added, such as DFA, DFZ, DFB, which allow the content of fatty acid methyl esters (FAMEs) up to 7% by weight), and with an addition of cold filter plugging point limitation.

Eighteen grades of marine fuels are distinguished in ISO 8217:

seven grades of distillate fuel–DMX, DMA, DMZ, DMB, DFA, DFZ, DFB;
eleven grades of residual fuel–RMA 10, RMB 30, RMD 80, RME 180, RMG 180, RMG 380, RMG 500, RMG 700, RMK 380, RMK 500, RMK 700.

Letters "D" and "R" in the grades' names indicate whether the fuel is distillate or residual, respectively, and the number indicates the kinematic viscosity.

The sulfur content of residual marine fuels is regulated by the International Convention for the Prevention of Pollution from Ships (MARPOL 73/78). The latest revision of the convention came into force on 1 January 2020; in accordance with Annex VI, the sulfur content in marine fuel is limited to 0.5% wt. Shipowners can circumvent this requirement if exhaust gas scrubbers are used on the ship. The International Maritime Organization (IMO) has also identified Special Environmental Control Areas (SECAs), in which the requirements for the operation of ships are even more stringent: starting from 1 January 2015, it is up to 0.1% wt. Such ECAs include the Baltic and North Seas, as well as some coastal areas of the United States and Canada [37].

In addition to international classification according to ISO 8217, there is also an exchange (trade) classification of marine fuels [38] (Table 1). The basic principle of labeling residual marine fuels (HFO, MFO, and IFO) on the exchange is the increasing content of distillate fractions in residual fuel, and the basic principle of labeling distillate marine fuels (MDO, MGO) is the reducing content of residual components in the fuel composition.

**Table 1.** The exchange (trade) classification of marine fuels and ISO 8217 standard compliance [34,36–38].

| International Trade Classification | ISO 8217 Compliance | Maximum Viscosity, cSt (at 50 °C for Residual Fuels and at 40 °C for Distillate Fuels) | Maximum Sulfur Content, % wt. |
|---|---|---|---|
| Heavy Fuel Oil (HFO) Heavy (residual) fuel, which either does not contain distillate fractions, or their proportion is minimal | RMG (RMK) 500/700 residual fuels | 500–700 | 3.5 |
| Medium Fuel Oil (MFO) Heavy (residual) fuel, which may contain a small proportion of distillate fractions | RMD, RME, RMG, RMK residual fuels | 80–500 | 3.5 |
| Intermediate Fuel Oil (IFO) Heavy (residual) fuel, which may contain a significant proportion of distillate fractions | RMD, RME, RMG, RMK residual fuels | 80–380 (500) | 3.5 |
| Marine Diesel Oil (MDO) Distillate fuel which may contain a small proportion of residual fractions | DMB distillate fuel and RMA, RMB residual fuels | ~11 (30) | 0.1–1.5 |
| Marine Gas Oil (MGO) Distillate fuel free of residual fractions | DMA and DMZ distillate fuels | 6 | 0.1–1.5 |
| HFO 0.1 or ECA Fuel (ULSFO) Residual, distillate, or mixed fuels with sulfur content of no more than 0.1% | Not standardized | - | 0.1 |
| HFO 0.5 (VLSFO) Residual, distillate, or mixed fuels with sulfur content of no more than 0.5% | Not standardized | - | 0.5 |

At present, due to the requirements of MARPOL, additions have been introduced for the marking of marine fuels with a sulfur content of less than 0.1% wt. used in ECA: low sulfur (LS) and/or sulfur content index (0.1), for example, LSMDO. The ultra-low-sulfur gas oil (ULSGO) indicates MGO with a sulfur content of not more than 0.1% wt. The ISO 8217 standard does not contain separate requirements for low-sulfur marine fuels, so it is not possible to make a clear correlation between ULSFO/VLSFO and ISO 8217 grades [39].

The number of bunkers produced by fuel with ultra-low-sulfur content ULSD is noticeably lower than this value for VLSFO, and the distribution by ports is fully correlated with the existing emission control zones in the Baltic Sea. Among the quality indicators, only density and viscosity are of research value, since other characteristics (such as sulfur, aluminum and silicon content, flow temperature) are within narrow limits that are included in the ranges required by regulatory documentation. Compared with VLSFO, the viscosity and density of ULSFO are noticeably lower, which is due to the fact that in order to achieve an ultra-low-sulfur content, residual refined components cannot be used, instead of which hydrotreated vacuum gas oil and low-sulfur diesel fractions with low viscosity and density values are mainly used. Singapore, the largest bunkering port in the world, produces VLSFO low-sulfur marine fuels of various compositions, but the most popular are compositions based on hydrotreated fuel oil (50–70%) and heavy cracker cycle oil (15–35%). Their average content in the released batches is 65% and 18%, respectively.

The exchange classification is currently one of the accurate ones, since it combines fuels of various types and classifications, and is quite clear for both manufacturers and consumers. Using this classification, it is possible to reliably assess production and market needs. In this regard, in this article, the assessment and analysis of data will be presented in accordance with the exchange classification: VLSFO-mixed low-sulfur fuels (sulfur content no more than 0.5% wt.); ULSFO-mixed low-sulfur fuels (sulfur content no more than 0.1% wt.); ULSGO-distillate marine fuels (sulfur content no more than 0.1% wt.); from high-sulfur fuels: MGO-distillate marine fuels (sulfur content no more than 1.5% wt.); HFO-residual marine fuels (sulfur content no more than 3.5% wt.) [40].

## 2. Marine Fuel Quality Indicator Analysis

*2.1. Methodology*

Information about real batches of marine fuels sold in the world's largest bunkering ports, collected and aggregated by the Lloyd's Register agency [41], was taken as the initial data for the analysis of quality indicators and calculation of the most popular compositions of VLSFO and ULSFO. Bunkering in the following world's leading ports was considered in this report:

- Asia: Singapore, Hong Kong;
- Western Europe: Amsterdam, Antwerp, Rotterdam;
- North America: Corpus Christi, Davant, Freeport, Gola, Pascagoula, Campeche;
- Middle East: Abu Dhabi, Dubai, Fujairah, Jabal Ali, Khor Fakkan, Hamriya, Sharjah;
- Russia: Kozmino, Murmansk, Nakhodka, Novorossiysk, Primorsk, St. Petersburg, Tuapse, Vanino, Vladivostok, Vostochny, Vysotsk, Port Bronka.

For the fuels covered in this article, the following quality indicators were studied: density, viscosity, cold flow properties, sulfur content, content of vanadium and sodium, aluminum plus silicon content.

The initial data consist of the batch fuel type (VLSFO, ULSFO, etc.) and grade (RMD80, RMG180, RMG380, etc.), port of bunkering, fuel supplier, date of bunkering, and fuel quality indicators mentioned above. These data were studied, processed based on the quality parameters, ports of bunkering, and fuel suppliers using statistical methods, and the results were provided in both the indicators (i.e., percentage of suppliers of certain fuel, its origins, etc.) and graphs.

The calculation of the most probable composition of a specific fuel batch was carried out with the help of linear programming optimization modeling. Using this type of modeling, the price of the resulting fuel was minimized while taking into account limits on its quality, both of which were calculated based on the fuel components and their share in the resulting fuel composition. These limits can be aggregated in the following equations:

$$\sum p_i x_i \rightarrow min,$$
$$b_{1L} \leq a_{11}x_1 + a_{12}x_2 + \cdots + a_{1n}x_n \leq b_{1H},$$
$$\ldots$$
$$b_{mL} \leq a_{m1}x_1 + a_{m2}x_2 + \cdots + a_{mn}x_n \leq b_{mH},$$

where $p_i$ is the price of the fuel component; $x_i$ is the share of the component in the composition; $b_{mL}$ and $b_{mH}$ are the low and high limits of the quality indicator in the composition; $a_{mn}$ is the quality indicator value for the fuel component.

*2.2. VLSFO Quality Analysis*

Data from more than 3600 VLSFO batches bunkered by 90 different companies from January–July 2021 in the chosen ports were analyzed [42]. The largest numbers of VLSFO bunkerings during the monitoring period were made by Equatorial Marine Fuel (6.2%), BP (5.4%), and Vitol (5.09%). Overall, there is no dominance of certain companies and the bunkerings are quite diversified. From a geographical point of view, Asia is greatly dominated since a key world port–Singapore–is located in the area.

The distribution of fuels bunkered in the considered world ports by their sulfur content is shown in Figure 1. As can be seen from the diagram, the vast majority of fuels have a sulfur content close to the limiting value of 0.5% wt., which indicates that manufacturers maximize fuel volumes by dilution of low-sulfur components with high-sulfur ones to the limit on sulfur content. The density distribution of the analyzed VLSFO batches bunkered in the world is also shown in Figure 1. The amount of VLSFO fuels increases uniformly with increasing density, reaching its peak in the area of 930–950 kg/m$^3$ and then decreasing again. This is consistent with the fact that the main component for the production of VLSFO in the world is hydrotreated atmospheric residue (with density of about 920–940 kg/m$^3$), and its key diluent is catalytic cracking heavy cycle oil (with density of about 1000–1050 kg/m$^3$).

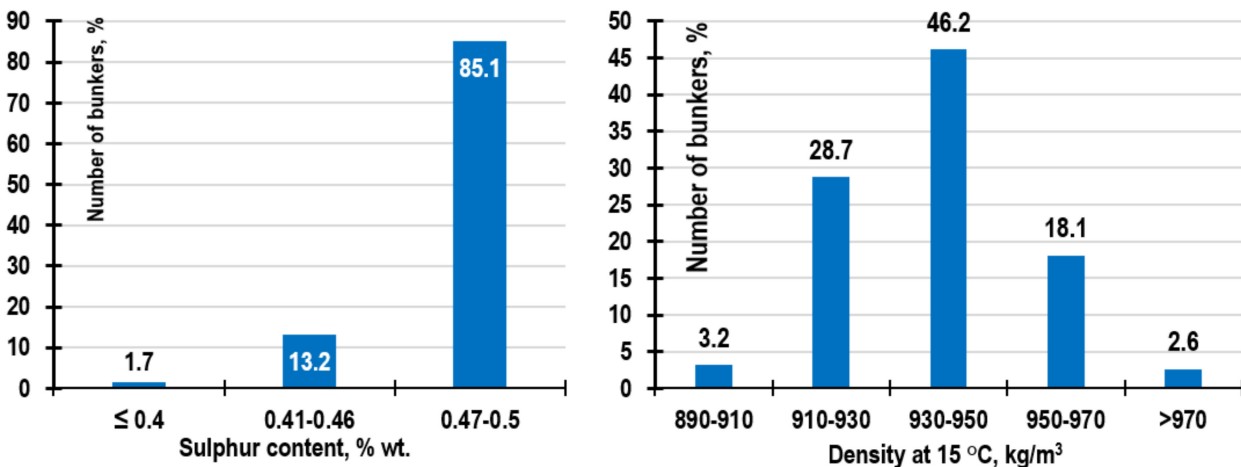

**Figure 1.** Sulphur content and density of VLSFO bunkered in world ports [43].

Figure 2 shows the viscosity characteristics of low-sulfur fuels bunkered in the chosen world ports. The graph is a normal distribution with a peak (most fuels) at a viscosity in the vicinity of 80 cSt. At the same time, there is a sharp decrease in the number of batches with a viscosity of less than 20 cSt, which is explained by the presence of heavy residual components in most VLSFOs, and of more than 180 cSt, which is associated with a low probability of obtaining such high-viscosity compounds since small amounts of low-sulfur diluents can significantly reduce the viscosity of the base residual fuel components.

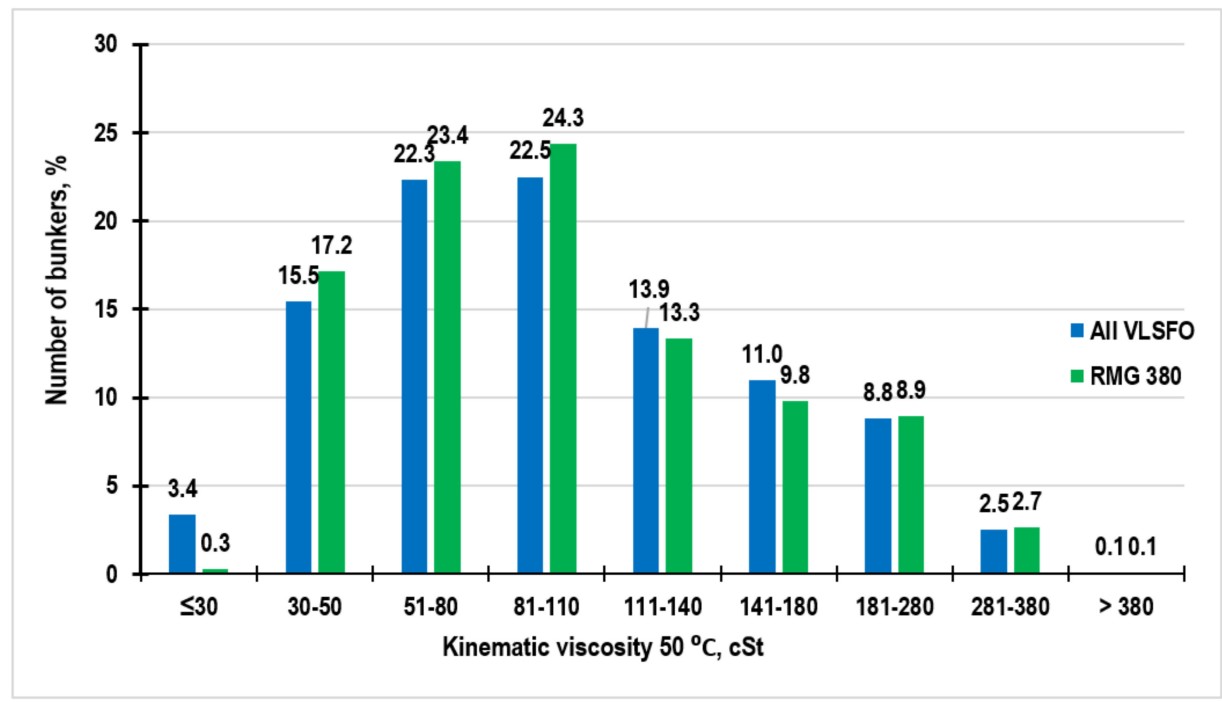

**Figure 2.** All VLSFO and RMG 380 kinematic viscosity (at 50 °C) [44].

The vast majority of analyzed bunkerings in the world were carried out with RMG 380 fuel, the regulatory requirements for which limit the kinematic viscosity at 50 °C to 380 cSt. However, at the same time, more than 85% of bunkered fuels of this grade had a viscosity of less than 180 cSt (Figure 2).

As shown in Figure 3, the limit on the total content of aluminum and silicon in fuels of various grades lies in the range of 25–60 ppm. Most fuels have a content of catalytic particles less than or slightly more than 40 ppm, significant excesses of this value were sporadic.

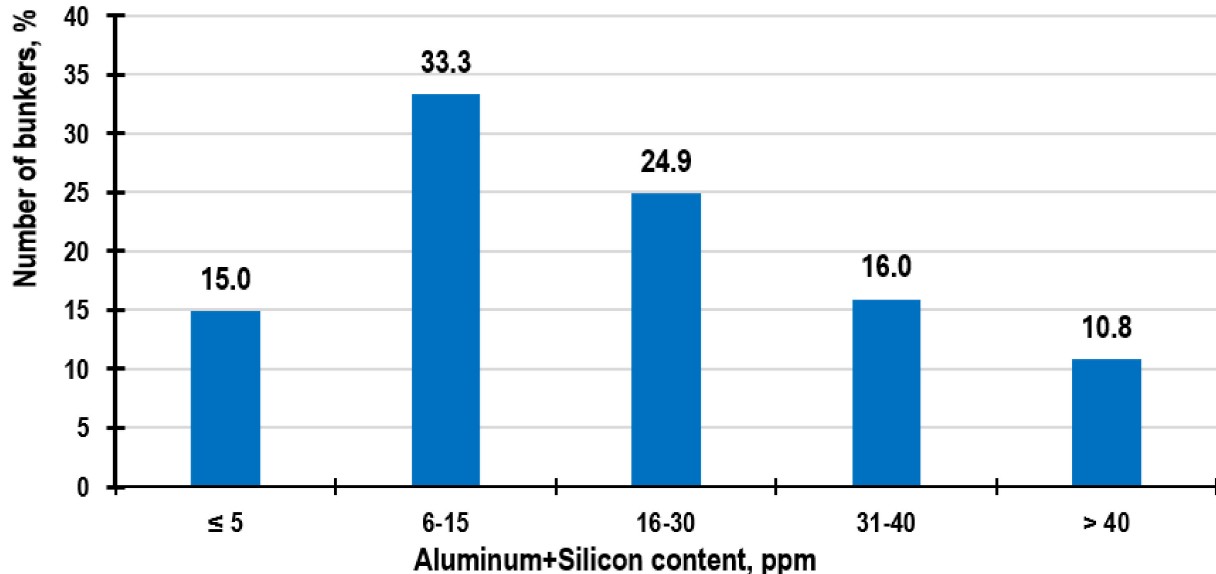

**Figure 3.** The total content of aluminum plus silicon in VLSFO [45].

A large amount of data provided by Lloyd's Register makes it possible to evaluate the correlations of various quality indicators of marine fuels. For example, the dependence of the content of aluminum plus silicon in the fuel on its kinematic viscosity is illustrated in Figure 4 (average total content of aluminum and silicon in the viscosity range based on bunkering data in the world). The key supplier of aluminum and silicon in marine fuels is catalytic cracker heavy cycle oil (HCO), so the maximum content of catalyst dust (that is essentially aluminum + silicon) is observed in the most probable viscosity range of 30–80 cSt. At lower viscosities (less than 30 cSt), HCO is usually not included in the fuel composition so as not to degrade its characteristics (primarily color). In the range of high-viscosity fuels (over 180 cSt), a decrease in the content of aluminosilicates is also observed, which is associated with a decrease in the total content of cracking products in them. At the same time, this value does not drop to zero even in the range of very high viscosities which is characteristic of practically pure hydrotreated atmospheric residue, which indicates the probable presence of catalyst dust in the residual components after hydrotreatment, by analogy with cracking products. Figure 5 shows a similar distribution of the average value of the content of aluminum plus silicon as a function of density.

The correlation of aluminum plus silicon content and fuel density does not have a clear peak but the plot shows an increase to a limit corresponding to a value of about 30 ppm. The maximum value being at the highest density range is also consistent with the fact that most of the catalyst particles enter the fuel as part of HCO, which, with a relatively low viscosity, is one of the most dense components of VLSFO and often determines the density of the entire composition.

By the content of vanadium and nickel in marine fuel, it is possible to judge the amount of residual components involved in it, since the absolute majority of oil metals are concentrated in oil residues [48,49]. The exception is hydrotreated residual fractions, in which the content of vanadium and nickel is about 10 times less than in non-hydrotreated raw materials. The dependence of the concentration of vanadium and nickel in the fuel on its viscosity is shown in Figure 6.

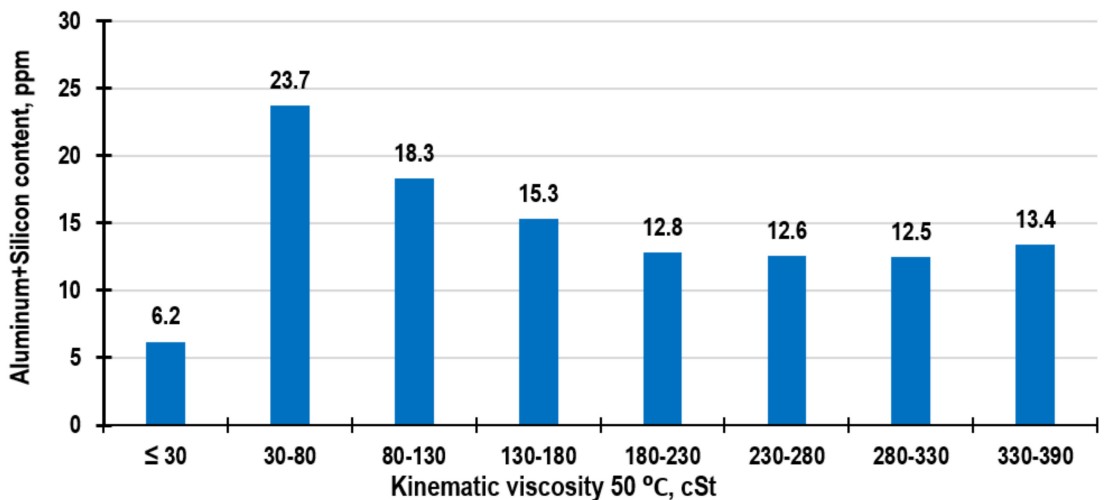

**Figure 4.** Aluminum plus silicon content depending on fuel viscosity [46].

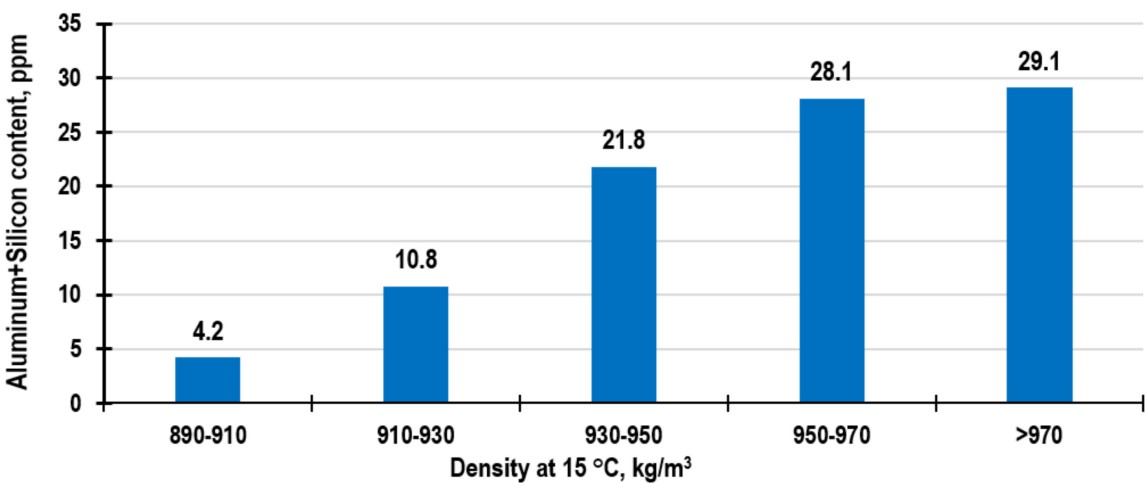

**Figure 5.** Aluminum plus silicon content depending on fuel density [47].

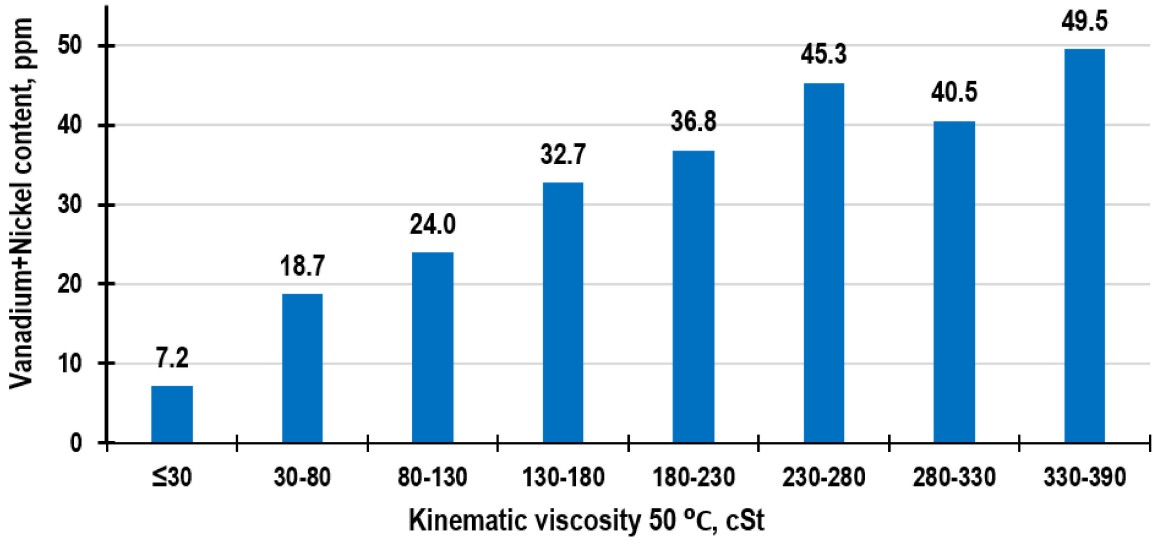

**Figure 6.** Dependence of the total vanadium and nickel content on the kinematic viscosity of the fuel in the averaged form [50].

It can be seen from the figures that the greater the kinematic viscosity of the fuel, the higher, other things being equal, the proportion of residual components in its composition, therefore, the dependence of the total content of these metals on the viscosity linearly increases to 230–280 cSt, after which the indicator stabilizes at around 40–50 ppm. A similar dependence for density is illustrated in Figure 7.

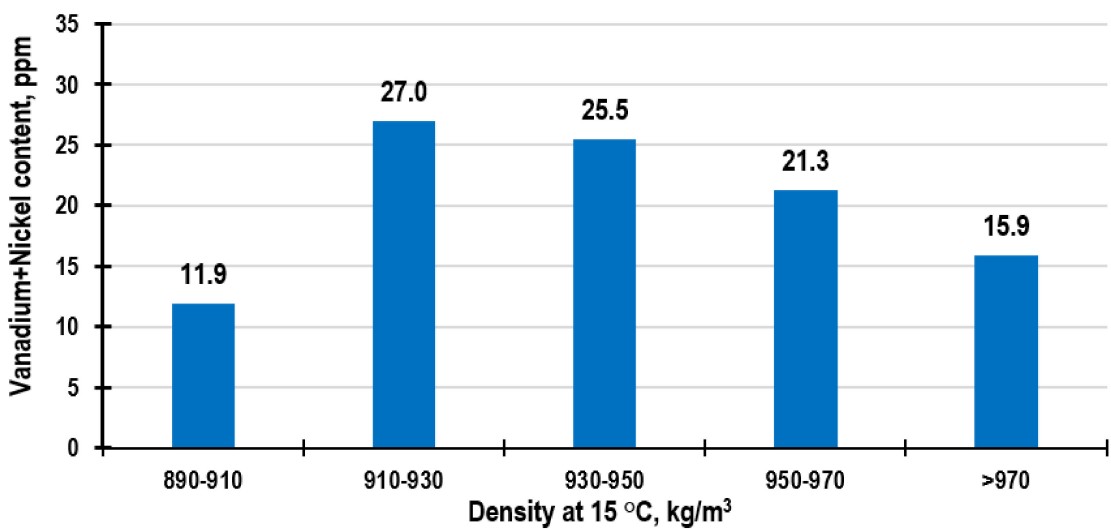

**Figure 7.** Dependence of the total vanadium and nickel content on the fuel density [50].

The graph shows a peak of vanadium and nickel content in the vicinity of 930 kg/m$^3$, which corresponds to the density of hydrotreated/hydrodesulfurized atmospheric residue (HDSAR). The decrease in the metal content after the peak is explained by the introduction of HCO into the fuel, which is a component that does not contain vanadium and nickel, but has a high density.

### 2.3. Analysis of ULSFO Quality Indicators

Fuel with ultra-low-sulfur content (ULSD, S $\leq$ 0.1%) is used in emission control areas (ECAs). Due to the fact that this territory is noticeably smaller than all the remaining regions, the number of bunkerings produced by ULSFO fuel is also significantly less and amounts to 151 batches. The distribution of kinematic viscosity of the analyzed ULSFO is shown in Figure 8.

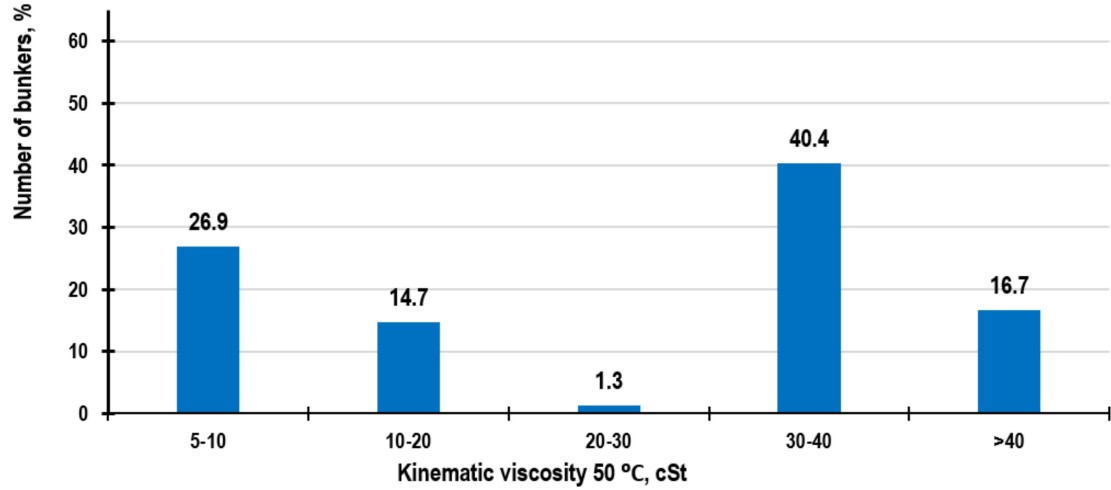

**Figure 8.** Kinematic viscosity of ULSFO bunkered in the world ports [50].

To analyze the diagram, it should be divided into two parts. The left region (up to 30 cSt) is characterized by a gradual decrease in the number of bunkered fuels with an increase in the viscosity value, which indicates that light distillates, such as diesel fuel (DF) and catalytic cracker light cycle oil (LCO), are used for the preparation of these fuels. The right part of the graph (viscosity more than 30 cSt) describes compositions, the main component of which is hydrotreated vacuum gas oil, which has a higher viscosity compared to DF and LCO. In general, the viscosity of ULSFO fuels is noticeably lower than VLSFO. This is due to the fact that in order to achieve an ultra-low-sulfur content, it is impossible to use residual refined components, the sulfur content of which remains at the level of 0.2–0.4% by weight, therefore, hydrotreated vacuum gas oil and low-sulfur diesel fractions are mainly used.

Similarly to viscosity, the density of ULSFO is significantly lower than the density of VLSFO for the same reason—a significantly higher content of light distillate fractions and hydrotreated vacuum gas oil in the fuel. Likewise, it is possible to observe the division of the histogram into two parts: the area of fuels based on hydrotreated diesel fuel (with a low density up to 900 kg/m$^3$) and the area of fuels based on HTVGO (with a higher density above 900 kg/m$^3$).

The vast majority of fuels with ultra-low-sulfur content has an extremely low content of catalytic particles: almost 90% of all bunkered fuels have a total concentration of aluminum and silicon of no more than 5 ppm, and the remaining 10%, from 5 to 10 ppm, which is associated with less use of HCO due to the high sulfur content in it, i.e., about 0.5%.

## 3. Analysis of Actual Marine Fuel Compositions

Out of all the characteristics of marine fuels, a number of properties were selected for calculating the most likely composition:

- density, viscosity, and sulfur content are the most important indicators of marine fuel, for which there are simple and fairly accurate calculation methods for the entire composition with known characteristics of individual parts and the values of which are well covered in the literature for key mixed components of marine fuels;
- the content of vanadium, nickel, aluminum, and silicon–these indicators cannot be used to accurately quantify the content of certain mixed components in fuels, but they serve as markers to determine the presence of individual fractions. Namely, the content of vanadium and nickel can be used to distinguish fuels that include residual components, while the sum of aluminum and silicon can help identify the presence of mixtures of HCO in the composition.

All potential components of marine low-sulfur fuel can be divided into two groups:

1. Components with higher density relative to viscosity.
2. Components with lower density relative to viscosity.

The reference components in this classification are the straight-run fractions: vacuum residue (VR), atmospheric residue (AR), vacuum gas oil (VGO), diesel fuel (DF). Graphically they can be represented in the form of Figure 9. As can be seen from the figure, straight-run fractions in the density double logarithm of viscosity coordinates fall on one straight line, and moving along which there is a smooth transition from vacuum gas oil (light VGO, heavy VGO, narrow fractions for lubricant production) to residual fractions (AR, topped residuum, VR). At the same time, non-straight-run fractions (i.e., produced with other units besides the crude distillation) deviate significantly from this straight line. The components obtained during the hydrocatalytic processes (hydrotreated AR and vacuum gas oil, hydrocracking residue) are above the straight line and have a lower density than the distilled component corresponding to them in viscosity, and the fractions obtained during the cracking processes (visbreaking residue, coking and catalytic cracker cycle oil) are under the straight line, and have higher density than the distilled component corresponding to them in viscosity, and the further they deviate from the "standard" viscosity–density characteristic (i.e., the straight line), the greater the degree of cracking in the processes of

their production. For example, catalytic cracker cycle oils, which contain vast amounts of aromatic hydrocarbons, are much further from the "standard" straight line than coking cycle oils, which, although they differ in structure and group composition from straight-run oil fractions, are still quite similar to them in viscosity–density properties.

Components lying on different sides of the straight line have different effects on the final properties of the composition of marine fuels: hydrotreated fractions reduce the density relative to viscosity. Cracking fractions, on the contrary, increase it. At the same time, different components from the same group–for example, heavy coking gas oil (HCGO) and heavy catalytic cracker cycle oil (HCO)–can play the same role in the final composition of VLSFO and are sometimes interchangeable [51]. Thus, marine fuel with certain viscosity–density characteristics can often be obtained by various combinations of secondary fractions, and it is not possible to reliably determine the true composition. In this case, when selecting the most likely composition, it is necessary to use additional markers–the content of metals and silicon—as well as take into account the total cost of the fuel.

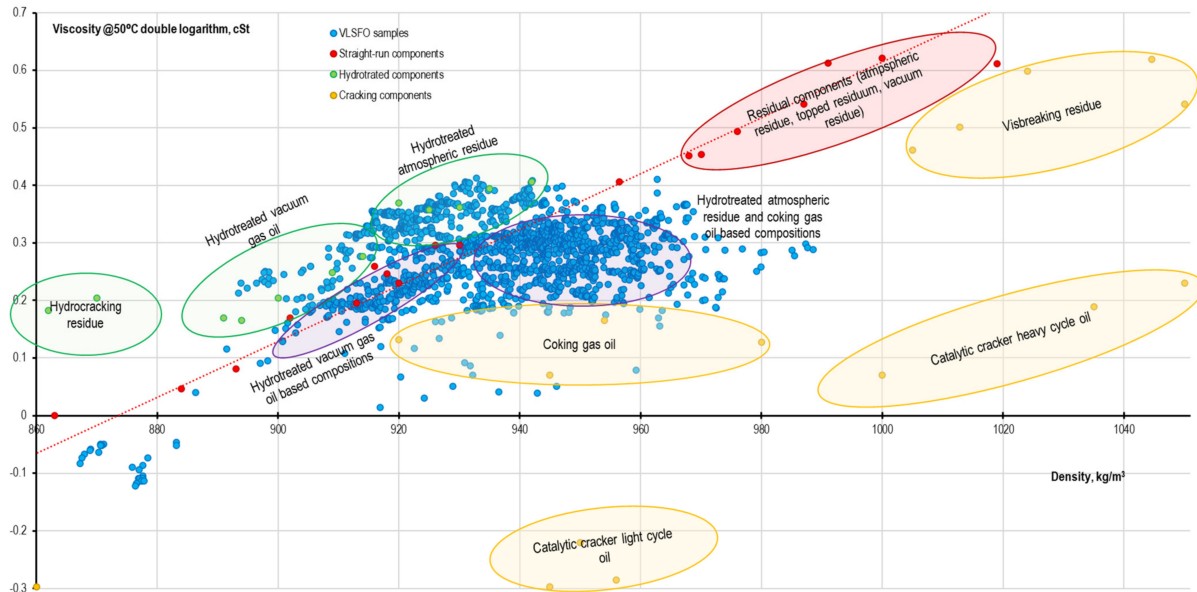

**Figure 9.** Graphical representation of viscosity–density properties of various components of marine fuels based on of real VLSFO samples [52]. Another important feature of the coordinates chosen for Figure 9 is the simplification of the perception of the process of blending various fractions and evaluating the properties of the final composition. So, if in the standard viscosity–density coordinates there is a non-linear (non-additive) change in the parameters during mixing, at the same time, when using a double logarithm, all fuel samples obtained by mixing two components with different properties will lie on one straight line connecting the points characterizing the pure components. For example, AR, which in principle is a mixture of VGO and VR, lies on a straight line connecting these fractions. This principle also justifies the presence of a direct "standard" viscosity–density characteristic consisting of oil fractions with different boiling points, but of the same nature.

According to Figure 9, it is also possible to conduct a surface analysis of compositions and the relative proportion of certain components in mixtures of low-sulfur marine fuels. Graphically, the results of the surface analysis are shown in Figure 9 in purple below the straight line. The first step to decode these batches would be to highlight the zones "Hydrotreated atmospheric residue" and "Hydrotreated vacuum gas oil". The batches located in them most likely largely consist of these components or their mixtures with the corresponding straight-run fractions (up to 30–40%). In addition, it is possible to distinguish the accumulation of batches in the zone of hydrotreated AR (density 920–935 kg/m$^3$), forming a straight line parallel to the "standard" one. Most likely, these batches are

mixtures of hydrotreated components (AR and vacuum gas oil) or target hydrogenates of the process of refining AR at different conversion and light component extraction rates.

Some of the batches are located in close proximity to the "standard" viscosity–density line in the area characteristic of straight-run vacuum gas oils: their density is 910–930 kg/m$^3$, which clearly indicates their qualitative composition, a significant proportion of which is occupied by straight-run vacuum gas oils with different boiling limits and sulfur content at the level of 0.4–0.6%.

The largest number of batches of marine fuels is located in the following zone: density 930–960 kg/m$^3$, viscosity 60–180 cSt (0.25–0.35 in the double logarithm), and significantly deviates from the "standard" straight line towards a higher density. Given the relatively small amount of oils, straight-run components of which (especially atmospheric residues) could form the basis of low-sulfur marine fuels and sufficiently high AR/VR hydrotreating capacities in fuel supplier countries (more details below), it can be assumed that these samples were obtained by mixing hydrotreated residual components and catalytic cracking gas oils [53]. Hydrotreated oil residues should be the base component, since only with their help is it possible to obtain a viscosity of 60–180 cSt without exceeding the sulfur limit. At the same time, catalytic cracking products are more likely to be the "dense" components, which is due to their low sulfur content and relatively high density. So, for example, coking gas oils can also make it possible to obtain batches with similar viscosity–density characteristics, but due to the high sulfur content in them, this fuel will not be low-sulfur. Additionally, the content of coking products in the final mixture is limited by the stability factors of the resulting fuel to oxidation. Due to their large share of olefin compounds, the addition of 20–30% of coking gas oils can lead to a loss of stability of the fuel. Catalytic cracking cycle oils consist mostly of aromatic hydrocarbons and do not affect the ability of the fuel to resist oxidation as much. At the same time, it is obvious that the disadvantages of coking products can be leveled by their deep hydrotreatment, however, in this case, the resulting fractions will lose their unique viscosity–density properties and will be indistinguishable from straight-run/hydrotreated components with the same boiling temperatures.

Further detailed analysis of batches of low-sulfur marine fuels was carried out in the context of individual fuel brands, supply ports, and companies that sell them [54]. The calculation of the most probable composition of a specific batch of fuel was carried out by linear programming by selecting the optimal qualitative and quantitative composition, which allows achieving the specified requirements for density, viscosity, and sulfur content while optimizing the cost of the resulting fuel.

*3.1. Analysis of VLSFO Fuel Compositions of RMG 380 Brand (Singapore)*

The analysis of the fuel compositions of low-sulfur marine fuels used in the world is carried out starting with the most popular grade—RMG 380—which accounts for about 90% of the total amount of VLSFO [55]. At the same time, the "true" RMG 380 fuels, the viscosity of which exceeds the 180 cSt limit for the RMG 180 brand, are only about 13%, that is, most marine fuel manufacturers unify the labeling of heavy VLSFO fuels under the RMG 380 brand. In this study, fuel batches are divided according to the brands indicated in the passports, regardless of the actual affiliation to less viscous brands.

In the world, the absolute leader in the number of batches of low-sulfur marine fuel is the port of Singapore—in the data taken for analysis, more than 75% of the batches were released from it. One of the largest suppliers of this port is the state-owned company Singamas Petroleum, whose share is about 6% of the total volume. Given the high share of this company and its connection with the Singapore public sector, the analysis of the most popular VLSFO compositions was started with it. The following fractions were used as the main mixed components in the calculations:

- the low-sulfur residual component is the target hydrogenate of fuel oil hydrotreating units (hereinafter referred to hydrotreated/hydrodesulfurized AR or HDSAR) with a

sulfur content of 0.5%, a density of 935 kg/m$^3$, and a viscosity at 50 °C of 300 cSt (for more details, see Section 3.2).

- Straight-run atmospheric residue (AR) is low-sulfur AR with a sulfur content of 1.0%, a density of 970 kg/m$^3$, and a viscosity of 700 cSt at 50 °C. With a higher sulfur content, the possibility of adding AR to low-sulfur fuel is severely limited, and this component is practically not involved in the modeling process.
- Catalytic cracker heavy cycle oil (HCO)–depending on the presence/absence of hydrotreating of the cracking feedstocks, the sulfur content in the HCO can vary widely, in this study the value of 0.5% is assumed, which facilitates the modeling of very different density samples. Other characteristics also strongly depend on the operating mode of the unit and the fractional composition of the final HCO, in this study the approximate characteristics of the 420 °C FBP fraction is taken: density 1050 kg/m$^3$ and viscosity at 50 °C 50 cSt.
- Heavy coking gas oil (HCGO)—as in the case of HCO, the properties of HCGO strongly depend on the parameters of the process and the feedstock. In this study, a low-sulfur VR coking product with the following characteristics was taken as a mixing component: 1.0% sulfur content, density 980 kg/m$^3$, viscosity 22 cSt at 50 °C.
- Hydrotreated vacuum gas oil (HTVGO) is a standard product of a catalytic cracker feedstock preparation unit: 0.1% wt. sulfur content, density 900 kg/m$^3$, viscosity 50 cSt at 50 °C.
- Diesel fuel (DF)—depending on the sulfur content in marine fuel, anything from straight-run diesel fuel to ultra-low-sulfur diesel fuel can be used. The properties are within the following limits: sulfur content 0.001–1.0%, density 828–840 kg/m$^3$, viscosity 2.0–2.4 cSt at 50 °C.

### 3.1.1. Singamas Petroleum

From here on, graphs will be used both for standard viscosity–density coordinates if the goal of the graph is to show general patterns in the distribution of samples according to the viscosity–density characteristic and logarithmic coordinates if the purpose is to analyze the principal compositions of batches [56,57].

The total number of bunkerings of this manufacturer exceeds 150 batches, therefore, to simplify subsequent modeling, it is necessary to reduce the scope of the data array. For this purpose, some groups of samples with similar properties were averaged. The results of simplifying the data array on Singamas Petroleum batches are shown in Figure 10 in red.

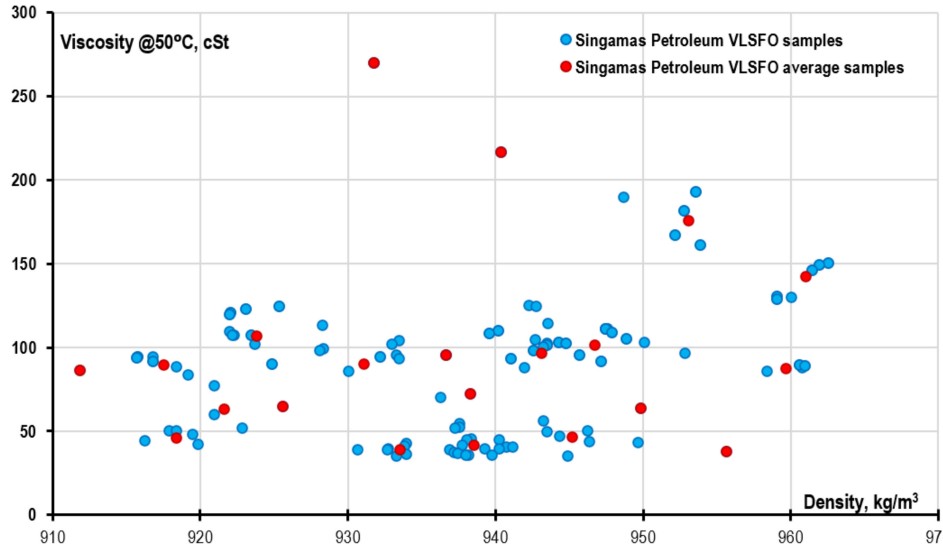

**Figure 10.** Singamas Petroleum average batches [58].

As can be seen from Figure 10, some unique batches (with very high viscosity) are only in red, since there were no other data with which they could be averaged. At the same time, most of the other samples were combined into joint points, which allowed reducing the total number of batches for subsequent modeling to 21. For these averaged batches, the optimal compositions were selected using the linear programming method, allowing us to achieve the specified viscosity–density properties and sulfur content. The characteristics of Singamas Petroleum batches and the results of modeling of probable fuel compositions are presented in Table 2.

**Table 2.** Averaged batches of Singamas Petroleum marine fuels and the results of modeling their composition [59].

| Sample No. | Density 15 °C, kg/m³ | Kinematic Viscosity at 50 °C, cSt | Sulfur Content, % | Vanadium Content, ppm | Aluminum and Silicon Content, ppm | Nickel Content, ppm | Calculated Content of Various Components in Fuel Samples | | | | | |
|---|---|---|---|---|---|---|---|---|---|---|---|---|
| | | | | | | | HDSAR | AR | HCO | HCGO | HTVGO | DF |
| 1 | 931.8 | 269.6 | 0.48 | 15.0 | 11.0 | 29.0 | 95 | 0 | 0 | 0 | 5 | 0 |
| 2 | 940.4 | 216.4 | 0.49 | 13.0 | 18.0 | 20.0 | 91 | 0 | 7 | 0 | 0 | 2 |
| 3 | 953.1 | 175.4 | 0.48 | 12.5 | 35.5 | 9.5 | 75 | 0 | 20 | 0 | 4 | 1 |
| 4 | 961.1 | 142.2 | 0.49 | 10.0 | 40.3 | 24.3 | 70 | 0 | 27 | 1 | 0 | 2 |
| 5 | 923.8 | 106.5 | 0.49 | 7.3 | 8.3 | 14.0 | 80 | 8 | 1 | 0 | 0 | 10 |
| 6 | 946.7 | 101.0 | 0.47 | 9.4 | 22.0 | 11.6 | 73 | 0 | 19 | 1 | 0 | 7 |
| 7 | 943.1 | 96.4 | 0.48 | 9.7 | 22.7 | 13.7 | 73 | 0 | 15 | 4 | 0 | 8 |
| 8 | 936.7 | 95.3 | 0.49 | 8.0 | 14.0 | 9.0 | 77 | 0 | 9 | 6 | 0 | 8 |
| 9 | 931.2 | 90.1 | 0.49 | 9.5 | 20.5 | 17.0 | 79 | 0 | 4 | 7 | 0 | 9 |
| 10 | 917.6 | 89.3 | 0.48 | 10.0 | 11.3 | 15.0 | 49 | 9 | 0 | 0 | 36 | 6 |
| 11 | 959.7 | 87.1 | 0.47 | 8.5 | 31.5 | 11.0 | 62 | 0 | 31 | 1 | 0 | 7 |
| 12 | 911.9 | 85.9 | 0.48 | 10.0 | 5.0 | 17.0 | 0 | 41 | 0 | 0 | 54 | 5 |
| 13 | 921.6 | 62.8 | 0.48 | 10.7 | 19.3 | 15.0 | 67 | 12 | 5 | 0 | 0 | 16 |
| 14 | 938.3 | 72.2 | 0.48 | 11.0 | 26.0 | 9.0 | 70 | 0 | 13 | 7 | 0 | 11 |
| 15 | 925.6 | 64.3 | 0.49 | 7.0 | 0.0 | 21.0 | 75 | 0 | 1 | 11 | 0 | 13 |
| 16 | 949.9 | 63.2 | 0.47 | 12.0 | 41.0 | 18.0 | 60 | 0 | 23 | 8 | 0 | 10 |
| 17 | 945.2 | 46.2 | 0.47 | 8.8 | 38.3 | 8.8 | 55 | 0 | 23 | 8 | 0 | 14 |
| 18 | 938.6 | 41.6 | 0.49 | 10.4 | 39.3 | 10.7 | 56 | 0 | 16 | 13 | 0 | 15 |
| 19 | 918.4 | 45.9 | 0.48 | 11.3 | 15.8 | 17.3 | 69 | 3 | 0 | 10 | 0 | 18 |
| 20 | 933.6 | 38.6 | 0.48 | 12.2 | 35.4 | 13.2 | 57 | 0 | 13 | 13 | 0 | 17 |
| 21 | 955.7 | 37.4 | 0.49 | 4.0 | 47.0 | 6.0 | 43 | 0 | 31 | 12 | 0 | 14 |

Summarizing the results obtained, we can say that the basis of the absolute majority of batches is hydrotreated fuel oil, the proportion of which ranges from 40–95%. The most important mixed component is heavy catalytic cracker cycle oil, the content of which in individual samples reaches 31%. Heavy coking gas oil and hydrotreated diesel fuel were also used in a large number of batches, but their ratio did not exceed 13% and 18%, respectively. Vacuum gas oil, on the contrary, is rarely used as a component of marine fuels: in only four averaged batches, two of which contained up to 5%. According to the content of individual components, it is possible to conditionally divide all batches into five types, graphically shown in Figures 11 and 12:

- type 1—HTFO–samples with a high content of hydrotreated fuel oil (80–95%), a small proportion of heavy gas oils of secondary processes (up to 7%), and an amount of diluent (diesel fuel) up to 15%;
- type 2—HTFO + HCO–samples based on hydrotreated fuel oil (60–75%) and heavy catalytic cracker cycle oil (15–35%) with a small fraction of diluent (up to 8%);
- type 3—HTFO + HCO + HCGO–samples based on hydrotreated fuel oil (55–80%) with an increased proportion of heavy coking gas oil (6–13%) and a low content of heavy catalytic cracker cycle oil (0–15%);
- type 4—HTVGO + HTFO–samples based on hydrotreated vacuum gas oil (45–60%) and fuel oil (30–50%) (a mixture of hydrotreated and straight-run);
- type 5—HTFO + HCO + DF–samples based on hydrotreated fuel oil (40–70%) and heavy catalytic cracker cycle oil (13–35%) with a small amount of coking gas oil (7–12%) and a high proportion of diluent (10–15%).

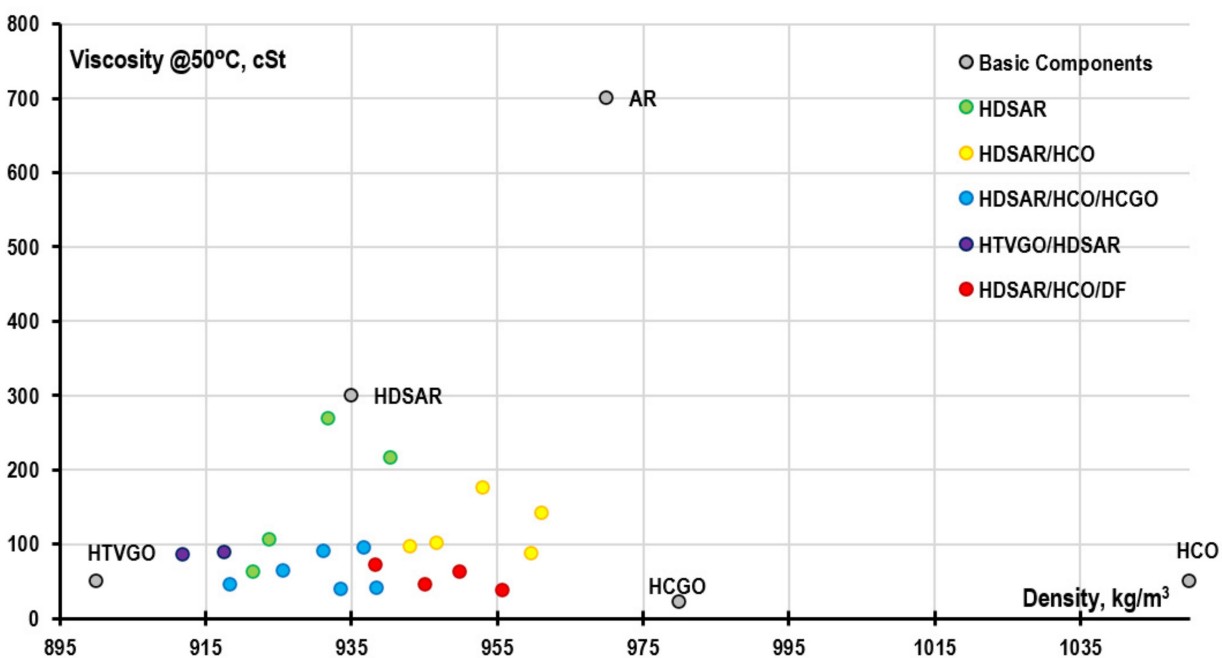

**Figure 11.** The most common types of fuel compositions in standard coordinates [60].

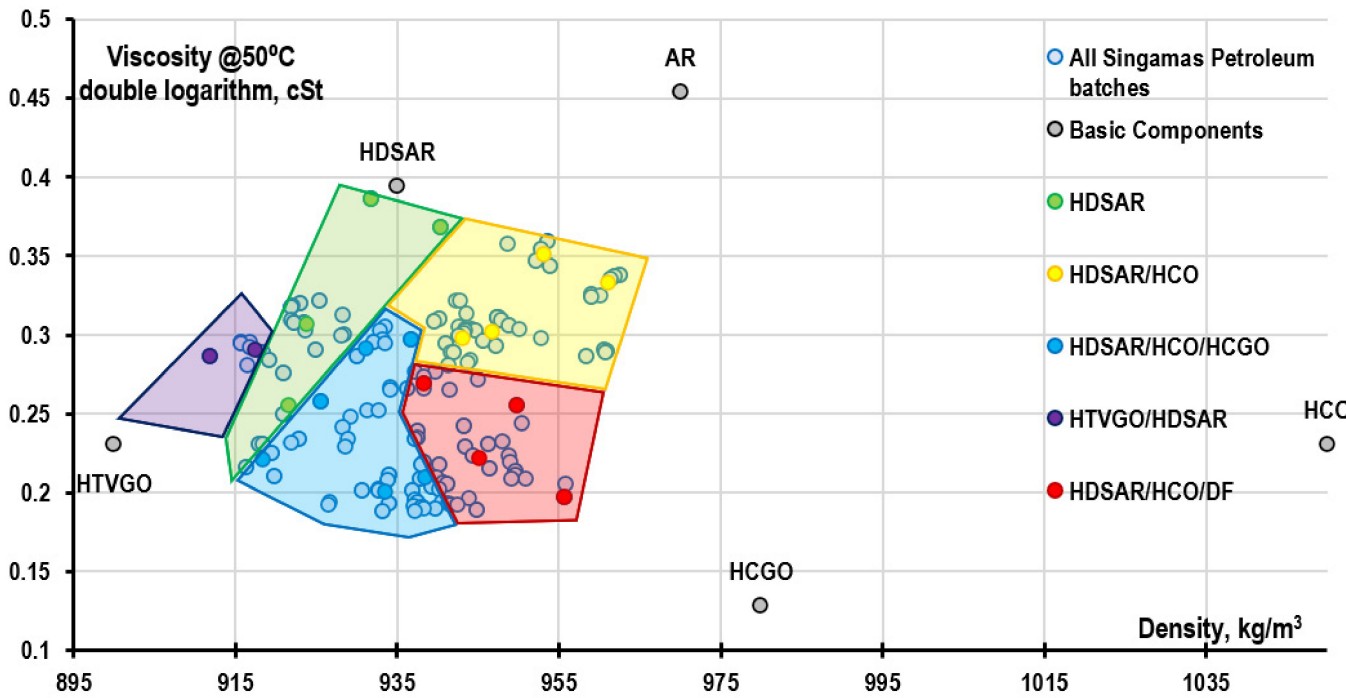

**Figure 12.** The most common types of fuel compositions and all Singamas Petroleum batches in logarithmic coordinates [60].

Figure 11 shows a general pattern of grouping samples according to their type (principal composition), nevertheless, the question remains unsolved as to why a certain fuel is most likely obtained by mixing the proposed components. For a more complete picture, Figure 6 is presented in logarithmic coordinates. When using the double logarithm of viscosity, the grouping of batches by type becomes much more understandable. For example, type 2 batches, consisting mainly of two components–hydrotreated fuel oil (AR) and HCO—are located between these components and the more they deviate from the

direct connecting point of the pure fractions, the more other components are in their composition, especially diesel fractions, which in the coordinates of the graph are located in the lower left corner outside the selected display area. Other fairly simple batches in terms of composition are samples classified as types 1 and 4. They consist of fuel oil and diesel fuel/vacuum gas oil and are located between these components in Figure 12. The remaining types are less "pure" and are mixtures of several components, therefore they are located in intermediate zones.

Thus, despite the similarity of the proposed compositions, it is possible to distinguish different types of basic fuel compositions that differ from each other in the set of components used and the proportion of their involvement. These compositions can be represented in the form of certain zones on the graph of the double logarithm of viscosity–density, and the batches falling into these zones can be characterized by a similar basic composition without additional calculations. At the same time, it is obvious that not all parties will strictly be within certain limits, some of them will be located on the border of two zones, that is, they will fit both types in composition, and some will go beyond the existing framework. For such samples, additional calculations must be carried out to establish the exact composition.

Figure 12 also shows all batches of Singamas Petroleum, distributed by zones of principal composition. This distribution is shown in Table 3.

**Table 3.** Distribution of Singamas Petroleum batches by type [61].

| Type No. | Approximate Composition of the Composition, % wt. | | | | | | Type Code Name | Number of Batches |
| | HDSAR | AR | HCO | HCGO | HTVGO | HTDF | | |
|---|---|---|---|---|---|---|---|---|
| 1 | 80–95 | 0–10 | 0–7 | <3 [1] | 0–5 | 0–15 | HTFO | 21 |
| 2 | 60–75 | <3 | 15–35 | 0–5 | 0–5 | 0–8 | HTFO + HCO | 41 |
| 3 | 55–80 | <3 | 0–15 | 6–15 | <3 | 5–20 | HTFO + HCO + HCGO | 50 |
| 4 | 30–60 | | <3 | <3 | 35–60 | 5–15 | HTVGO + HTFO | 7 |
| 5 | 40–70 | <3 | 10–35 | 5–12 | <3 | 10–15 | HTFO + HCO + DF | 31 |

[1] From here on, "<3" is used as an analogue of zero content, that is, if the calculations did not show the presence of any component, this does not mean that it cannot be included in the composition in an extremely small amount (less than 3%).

The most popular fuel compositions are types 2, 3, 5, from which it can be assumed that in reality all these compositions are obtained using the same technology, slightly varying the ratio of components. At the same time, types 1 and 4 are unique and are obtained using a different mixing technology with minimal use of heavy coking and catalytic cracker cycle oils.

To simplify perception, a code name was compiled for each type of composition, based on the components that occupy a large proportion or are a distinctive feature of the composition.

Using the example of Singamas Petroleum batches, a method for calculating and deciphering basic fuel compositions of low-sulfur fuels from various manufacturers is shown. Further simulation results will be presented in a simplified form without a large number of explanations and examples.

3.1.2. British Petroleum

BP is one of the largest suppliers of marine low-sulfur fuels in the world and in particular in Singapore, in whose ports the Lloyd's Register agency has recorded more than 170 bunkering operations over the past 6 months [62]. Graphically, the fuel batch data are presented in Figure 13.

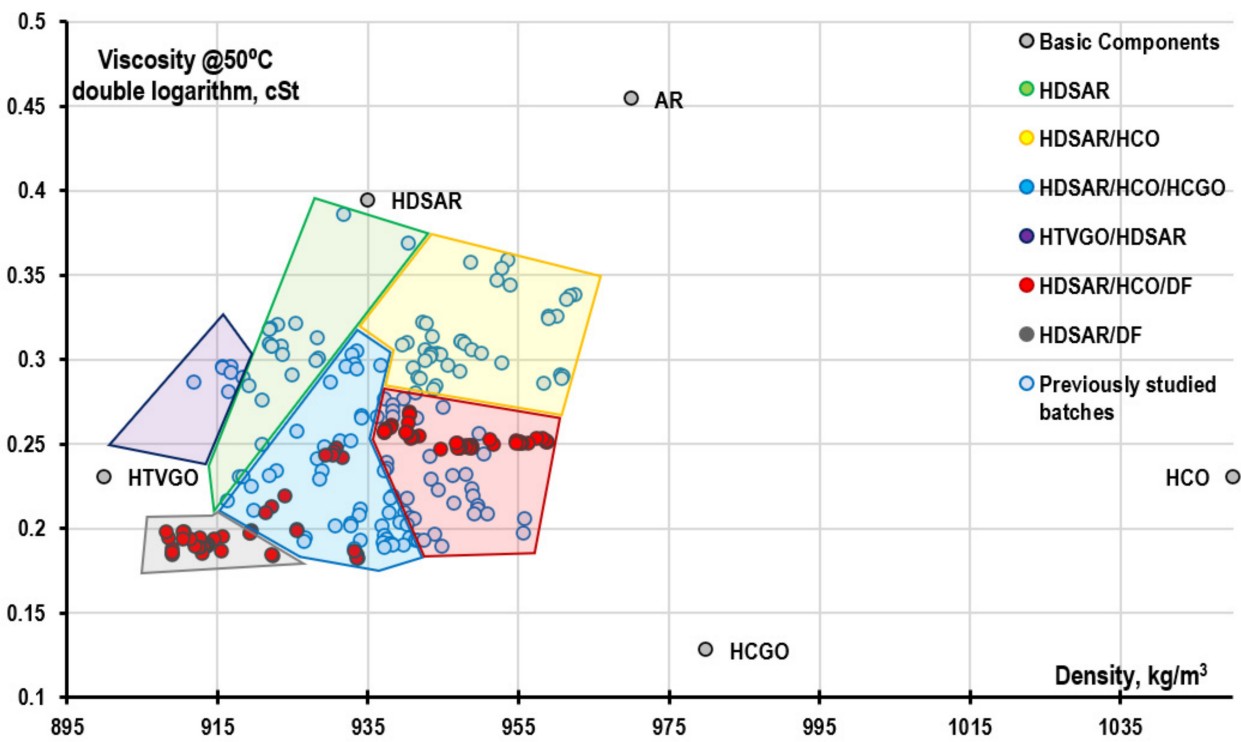

**Figure 13.** BP batches in logarithmic coordinates [63].

A distinctive feature of the BP marine fuel production technology is a small viscosity spread (30–75 cSt) with a very large variability in density (905–960 kg/m$^3$). The results of modeling the averaged samples presented in Table 4 illustrate that the batches included in any area in Figure 13 mainly correspond to them in principle composition, but may also additionally include small amounts of hydrotreated vacuum gas oil. Of particular interest are samples with a density below 925 kg/m$^3$.

**Table 4.** Averaged BP batches and results of modeling their composition [64].

| No. | Density 15 °C, kg/m$^3$ | Kinematic Viscosity, cSt | Sulfur Content, % | V, ppm | Al + Si, ppm | Ni, ppm | Calculated Content of Components, % wt. | | | | |
| --- | --- | --- | --- | --- | --- | --- | --- | --- | --- | --- | --- |
| | | | | | | | HDSAR | HCO | HCGO | HTVGO | DF |
| 1 | 956.5 | 60.5 | 0.48 | 6.6 | 49.1 | 6.2 | 58 | 25 | 0 | 5 | 11 |
| 2 | 948.0 | 59.1 | 0.48 | 7.2 | 48.3 | 11.6 | 56 | 28 | 0 | 5 | 11 |
| 3 | 939.3 | 65.1 | 0.48 | 8.3 | 32.6 | 5.3 | 62 | 20 | 0 | 7 | 11 |
| 4 | 933.5 | 33.6 | 0.49 | 10.2 | 44.8 | 12.0 | 55 | 19 | 7 | 0 | 20 |
| 5 | 930.6 | 56.7 | 0.49 | 7.2 | 39.6 | 14.4 | 70 | 14 | 0 | 1 | 15 |
| 6 | 922.2 | 38.6 | 0.49 | 7.4 | 24.3 | 8.4 | 66 | 9 | 5 | 0 | 20 |
| 7 | 912.0 | 35.5 | 0.49 | 8.2 | 18.4 | 13.0 | 70 | 1 | 6 | 0 | 23 |

The averaged sample 6, summarizing the indicators of 16 batches with a density of 919–925 kg/m$^3$, is very close to type 3 in its composition, which is also confirmed graphically—half of the batches fall into the blue-highlighted zone—so all bunkering data were assigned to this type.

The average sample 7, representing 39 batches with a density from 908 kg/m$^3$ to 916 kg/m$^3$, is also very close to type 3 in composition, but differs in that the total content of heavy gas oils in it does not exceed 10%, and the proportion of diesel fuel, on the contrary, is quite large (more than 20%).

Thus, the basic technology for producing BP mixed fuel is similar to that described earlier by Singamas Petroleum and consists primarily in the use of two components:

hydrotreated fuel oil and heavy cracker cycle oil. However, in addition to the compositions formulated earlier, another type is also characteristic of BP:

- type 6—HTFO + DF–based on hydrotreated fuel oil (60–75%) with a low content of heavy gas oils of secondary processes (up to 10%) and an increased proportion of diesel fuel (15–30%).

When analyzing Figure 13, the idea may also arise that BP compositions are based on a mixture of hydrotreated vacuum gas oil and HCO, since most of the batches are located in close proximity to the straight line connecting these components. However, a more detailed analysis of this hypothesis shows that all fuel samples have a sufficiently high content of vanadium and nickel—in the amount of 10–25 ppm—which indicates the presence of a residual component in them. If this indicator were provided by the addition of a small amount of straight-run fuel oil/tar, it would fluctuate within much larger limits, involving dropping to near zero values, and would somehow correlate with the density of samples, which is not observed in reality. Consequently, it is more likely that a large amount of hydrotreated fuel oil is present as a residual component, which provides approximately the same metal content in all batches and is the basis of the mixture. In addition, the very process of modeling mixtures of hydrotreated vacuum gas oil and heavy cracker cycle oil revealed the impossibility of obtaining most of the averaged samples without using a significant amount of hydrotreated fuel oil.

The distribution of British Petroleum batches by principal types of compositions is shown in Table 4.

### 3.1.3. Eng Hua Company

During the period under review, this company carried out 113 deliveries of low-sulfur marine fuel. Graphically, the viscosity–density properties of these batches are shown in Figure 14.

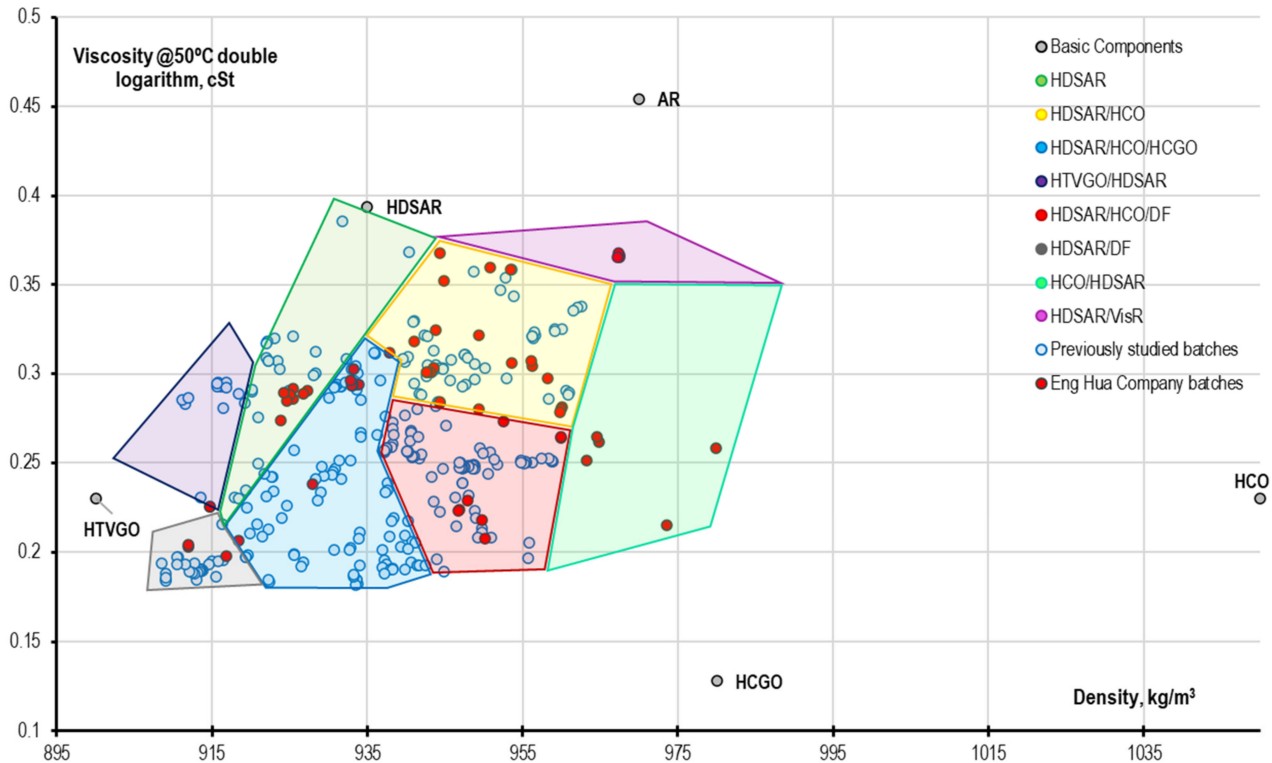

**Figure 14.** Eng Hua Company batches in logarithmic coordinates [65].

A distinctive feature of the batches of this company is the presence of samples with a very high density (above 960 kg/m$^3$) and different viscosity values: 40–70 cSt and 170–220 cSt, so they were subjected to detailed modeling. The simulation results are presented in Table 5.

**Table 5.** Averaged Eng Hua Company batches and simulation results [66].

| No. | Density 15 °C, kg/m$^3$ | Kinematic Viscosity, cSt | Sulfur Content, % | V, ppm | Al + Si, ppm | Ni, ppm | Calculated Content of Components, % wt. | | | | | |
|-----|-----|-----|-----|-----|-----|-----|-----|-----|-----|-----|-----|-----|
| | | | | | | | HDSAR | VGO | HCO | HCGO | HTVGO | DF |
| 1 | 979.9 | 65.1 | 0.46 | 8.0 | 29.0 | 7.0 | 37 | 0 | 50 | 0 | 7 | 5 |
| 2 | 973.5 | 43.8 | 0.50 | 3.0 | 44.0 | 2.0 | 38 | 0 | 50 | 0 | 0 | 12 |
| 3 | 964.2 | 65.7 | 0.48 | 6.0 | 22.3 | 8.7 | 53 | 0 | 38 | 0 | 1 | 9 |
| 4 | 967.4 | 211.3 | 0.49 | 9.3 | 9.0 | 13.8 | 69 | 6 | 24 | 0 | 0 | 0 |
| 5 | 948.4 | 194.8 | 0.48 | 17.0 | 32.5 | 17.5 | 78 | 0 | 15 | 0 | 7 | 0 |

Based on the simulation results, it is possible to identify new types of compositions peculiar to Eng Hua Company:

- type 7—HCO + HDSAR—samples 1–3 are similar in composition to type 2 "HDSAR +HCO", but have a significantly increased content of HCO, since only with its help is it possible to obtain components with such a high density at a relatively low viscosity. These samples were placed in a separate type, the name of which reflects the dominant position of cracking products in the composition;
- type 8—HDSAR + VR—sample 4 has similar density characteristics to samples 1–3, but significantly higher viscosity. Moreover, it is so high that in order to achieve it, there are not enough available components and it is necessary to introduce an additional high-density and high-viscosity fraction, for which the visbreaking residue (VisR) obtained from low-sulfur tar was chosen, with a density of 1050 kg/m$^3$, a viscosity of 50 cSt, and a sulfur content of 1.5% wt.

The batches forming the averaged sample 5, although they have somewhat atypical viscosity–density properties, fit the existing framework for type 2 "HDSAR +HCO" and were assigned to it. The distribution of Eng Hua Company batches by principal types of compositions is shown in Table 5

### 3.1.4. Minerva Bunkers

During the period under review, this company delivered 119 batches of low-sulfur marine fuel. Graphically, the viscosity–density properties of these batches are shown in Figure 15.

Among the batches of this company, only samples with the smallest viscosity are of interest, clearly falling out of the existing zones of principle types. Additional calculations were carried out for them in order to establish the intended composition. The results are presented in Table 6.

This averaged sample consists of two key components: hydrotreated vacuum gas oil and heavy coking gas oil, therefore it cannot be attributed to existing types and will form its own–type 9 "HTVGO + HCGO" (already presented in Figure 15). The distribution of Minerva Bunkers parties by principal types of compositions is shown in Table 6.

### 3.1.5. Other Manufacturers

Further detailed analysis of all large bunkering companies in Singapore did not reveal significant differences in the types of fuel compositions used, therefore, in order to reduce the total amount of work, only a generalizing characteristic of the most popular basic compositions of low-sulfur marine fuels for different manufacturers will be provided (Table 7).

Therefore, the most popular fuel compositions of low-sulfur marine fuels in the port of Singapore are compositions that include a hydrotreated residual component (fuel oil) and heavy cracking gas oil in various concentrations. The average content of refined AR in fuel samples is about 64%, heavy cracking gas oil–18%, the remaining components are much less frequently used and therefore occupy a smaller share: diesel fuel—9%, hydrotreated vacuum gas oil–5%, heavy coking gas oil–4%, straight-run AR/VR residue–up to 1%.

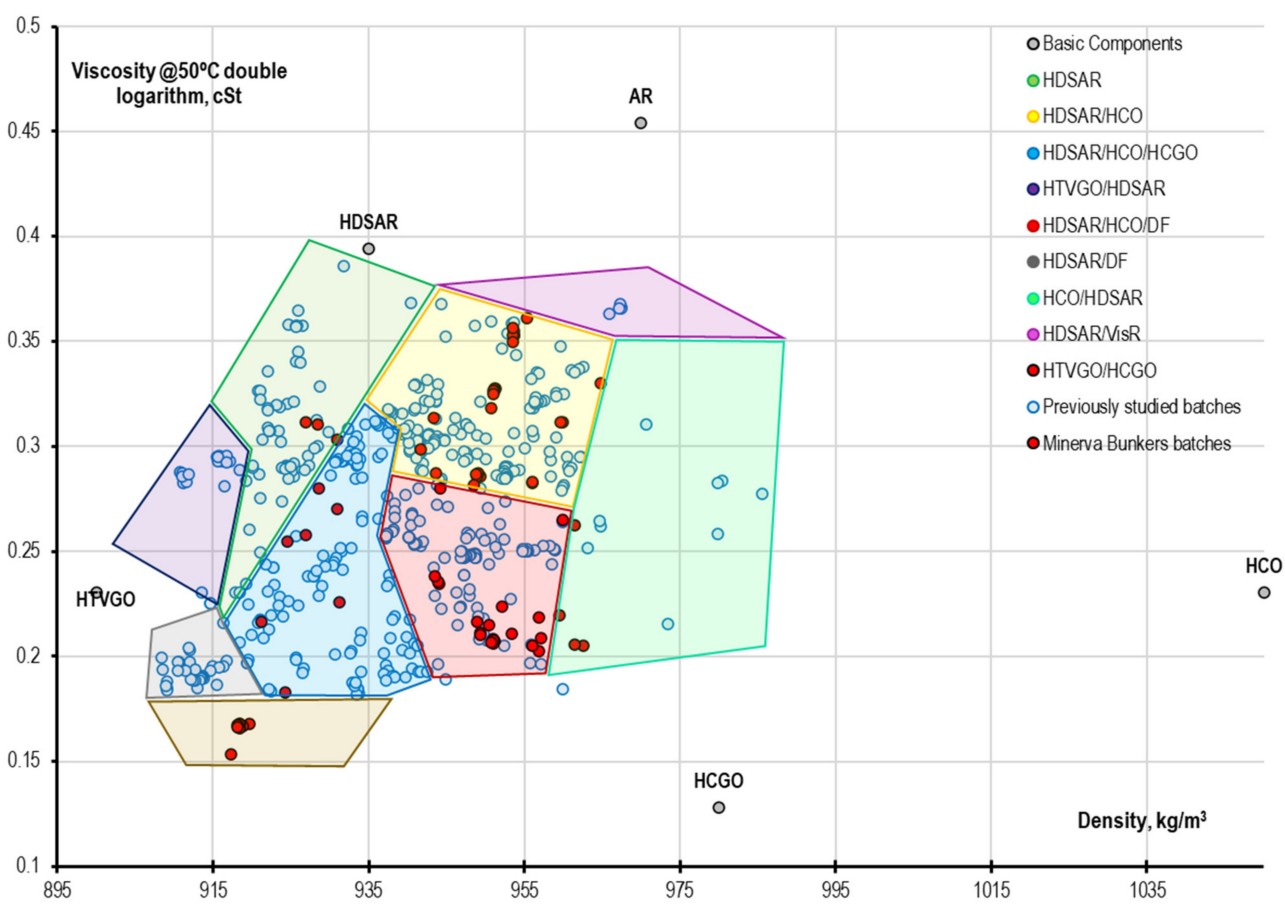

**Figure 15.** Minerva Bunkers batches in logarithmic coordinates [67].

**Table 6.** Averaged Minerva Bunkers batches and simulation results [68].

| No. | Density 15 °C, kg/m³ | Kinematic Viscosity, cSt | Sulfur Content, % | V, ppm | Al + Si, ppm | Ni, ppm | Calculated Content of Components, % wt. | | | | | |
|---|---|---|---|---|---|---|---|---|---|---|---|---|
| | | | | | | | HDSAR | AR | HCO | HCGO | HTVGO | DF |
| 1 | 918.1 | 29.1 | 0.49 | 7.6 | 5.3 | 15.9 | 0 | 4 | 0 | 29 | 60 | 7 |

**Table 7.** Distribution of RMG 380 shipments in the port of Singapore by type [69].

| Composition Number (Type) | | 1 | 2 | 3 | 4 | 5 | 6 | 7 | 8 | 9 |
|---|---|---|---|---|---|---|---|---|---|---|
| Code Name | | HDSAR | HDSAR + HCO | HDSAR + HCO + HCGO | HTVGO+ HDSAR | HDSAR + HCO + DF | HDSAR + DF | HCO + HDSAR | HDSAR + VR | HTVGO + HCGO |
| Composition of the composition, % by wt. | HDSAR | 80–95 | 60–75 | 55–80 | 30–60 | 40–70 | 60–75 | 35–55 | 60–75 | 0–7 |
| | AR/VR | 0–10 | <3 | <3 | | <3 | <3 | <3 | 0–10 | |
| | HCO | 0–7 | 15–35 | 0–20 | <3 | 10–35 | 0–10 | 35–50 | 0–25 | <3 |
| | HCGO | <3 | 0–5 | 0–15 | 0–10 | 0–12 | | 0–5 | <3 | 15–30 |
| | HTVGO | 0–20 | 0–10 | <3 | 35–60 | 0–7 | <3 | 0–10 | 0–10 | 50–70 |
| | HTDF | 0–15 | 0–8 | 5–20 | 5–15 | 10–15 | 15–30 | 0–15 | <3 | 0–10 |
| The number of batches assigned to certain types | | | | | | | | | | |
| BP | | - | - | 29 | - | 87 | 57 | - | - | - |
| Cathay Marine | | 7 | 25 | 27 | 9 | 9 | 3 | - | - | - |
| Eng Hua Company | | 16 | 49 | 12 | 2 | 12 | 6 | 10 | 6 | - |
| Equatorial Marine Fuel | | 50 | 67 | 46 | 11 | 24 | 4 | 4 | 4 | - |
| ExxonMobil | | - | 93 | - | - | - | - | 5 | - | - |
| Global Energy Trading | | 11 | 50 | 2 | - | 15 | 4 | 2 | 6 | - |
| Golden Island | | 15 | 72 | - | - | - | - | - | - | - |
| Hong Lam Marine | | 12 | 38 | 60 | - | 52 | 12 | 2 | - | - |
| Minerva Bunkers Pte Ltd. | | 6 | 42 | 17 | - | 38 | - | 4 | - | 12 |
| Petro China | | - | 29 | 13 | - | 4 | - | 10 | - | - |
| Sentek Marine | | 6 | 34 | 36 | 6 | 11 | 13 | 4 | - | - |
| Shell | | 8 | 53 | 17 | - | 58 | - | 11 | - | - |
| Singamas Petroleum | | 21 | 41 | 50 | 7 | 31 | - | - | - | - |
| SK Energy | | 2 | 28 | 27 | - | 44 | - | 15 | - | - |
| TFG Marine | | 6 | 8 | 31 | - | 2 | 38 | - | - | - |
| Vitol | | 34 | 65 | 14 | - | 48 | - | 9 | - | - |
| Total: | | 194 | 694 | 352 | 35 | 348 | 80 | 76 | 16 | 12 |
| % of the total | | 10.7 | 38.4 | 19.5 | 1.9 | 19.3 | 4.4 | 4.2 | 0.9 | 0.7 |

*3.2. Origin of the Low-Sulfur Residual Component*

Previously, a "low-sulfur residual component" was declared as the main component of low-sulfur marine fuels, which was further referred to as hydrotreated fuel oil [70]. This fraction made it possible to calculate the compositions of compositions without answering the question about the true nature of the basic residual component of individual batches of marine fuels, since it is often not possible to distinguish two VLSFO samples, one of which utilized straight-run low-sulfur fuel oil (up to 1.0–1.5% sulfur), and the other hydrotreated. Even the indicators noted earlier–the contents of vanadium and nickel–are not able to clearly distinguish between these two samples, since often extremely low-sulfur oil can also contain very low amounts of metals, and the degree of demetallization during hydrotreating of fuel oil can vary significantly depending on the specific raw materials and cleaning technology. Consequently, the value of the sulfur content in the "low-sulfur residual component" was chosen experimentally at the level of 0.5–0.6%, which makes it possible to simulate all the investigated batches under equal conditions, that is, without reducing or increasing the sulfur content in individual fractions for individual manufacturers due to the inability to obtain the declared properties of fuels.

After considering all the batches of the port of Singapore, one can return to the question of the composition of the "low-sulfur residual component" and try to calculate the ratio of hydrotreated and straight-run fuel oil in it. A mixture of hydrotreated (80–90%) and straight-run fuel oil (10–15%) with a sulfur content of no more than 2.0–2.5% is used as the base low-sulfur component of marine fuels in the ports of Singapore and the Middle East. In other Asian ports (for example, Hong Kong), popular fuel compositions roughly resemble Singapore ones, but the share of heavy cracking gas oil in the average batch is even higher–about 20%.

Singapore is one of the largest oil refining centers, but oil production itself is not carried out in the country, and the absolute majority of it is imported. The largest number of imports are sent from the Middle East (Saudi Arabia, Kuwait, UAE, Iran), thus, data on the quality of oil and its production volumes for the Middle East will be utilized as a starting point for calculating the parameters of raw materials used in Singapore.

As noted earlier, the content of metals (vanadium and nickel) is an indicator of the presence of residual components in the mixture, but it is impossible to use this indicator to calculate the relative concentration of straight-run AR in the context of individual batches due to the high spread of its values for specific oils/fuel oils. However, one can try to use the metal content specifically for quantitative assessment, when considering the overall picture (all the batches of Singapore as a whole). Table 8 presents the typical contents of metals and sulfur for Arab oils.

**Table 8.** The content of metals and sulfur in various grades of commercial oils of the Middle East [71].

| Impurity | The Content of Metals and Sulfur in Grades of Commercial Oil, mg/kg and wt.% | | | | |
|---|---|---|---|---|---|
| | Arabian Heavy | Iranian Heavy | Iranian Light | Kuwait | Arabian Light |
| Vanadium | 69.8 | 68.2 | 55.2 | 32.9 | 23.7 |
| Nickel | 22.3 | 21.4 | 17.0 | 9.6 | 4.6 |
| Total metals | 92.1 | 89.6 | 72.2 | 42.5 | 28.3 |
| Sulfur | 3.0 | 2.3 | 1.5 | 2.6 | 1.9 |

Taking into account the relative amounts of extracted oils of each grade (Figure 16) [72], it is possible to calculate the average metal content in oil coming from the Middle East: vanadium–45.3 ppm; nickel—13.2 ppm; a total content of 58.5 ppm. In the composition of oil, metals are concentrated exclusively in residual fractions, thus, neglecting their content in all distillate products, it is possible to roughly estimate the proportion of metals in the average fuel oil at the Singapore refinery, which is equal to 117 ppm. In this case, in the HDSAR that has undergone demetallization, the metal content will decrease to 10–15% of the original, that is, to 12–18 ppm.

It follows from the conclusions of the last section that the average content of the "low-sulfur residual component" in the studied batches is 64%. At the same time, the average total content of vanadium and nickel in them is 23.2 ppm, which allows us to estimate the proportion of metals in the most residual component–35.7 ppm. To achieve this value by blending "average" straight-run and HDSAR, it is required to maintain a ratio of 20% to 80%, respectively. Taking into account the average sulfur content in commercial grades of oil and their blends entering the Singapore refinery, it is possible to calculate the amount of sulfur in the "low-sulfur residual component" at this fraction ratio. The average sulfur content in the oil blend is 2.25%, in the fuel oil of these oils it is 3.5–4.0%, in the AR after hydrotreating it is 0.3%, respectively, and in the residual component, it is about 1%. It is necessary to use either less straight-run AR, or select relatively low-sulfur fuel oil (1.5–2.5%) to obtain a "low-sulfur residual component" with the declared properties (sulfur content 0.5–0.6%). Both options look quite likely, thus, an intermediate option was chosen as the optimal composition of the "low-sulfur residual component", which would describe the real situation as closely as possible: 80–95% HDSAR (the target hydrogenate of the vacuum residue hydrotreating/hydrodesulfurizing unit can also be used, subject to the fractional composition of the AR, that is, without excessive extraction of light fractions) and 10–15% straight-run low-sulfur fuel oil (up to 2.0–2.5% sulfur) or 5–10% of the typical Middle Eastern oil sulfur fuel oil (3.5–4.0% sulfur). Further in the text, as in Section 3.2, to simplify writing and perception, this mixture will be called "HDSAR" or less often "low-sulfur residual component".

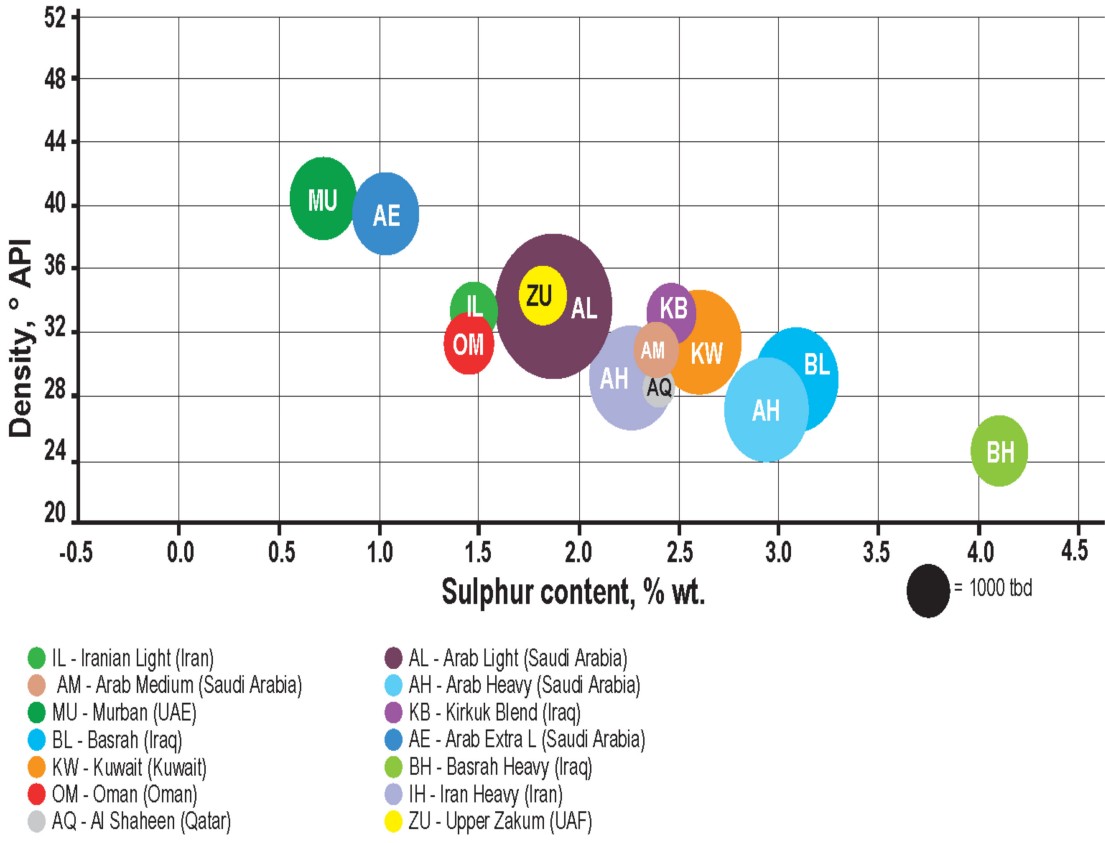

**Figure 16.** Quantity and quality of Middle East oil [73].

Given the geography of oil supplies to Singapore, we can expect a similar composition of the low-sulfur residual component in the batches of Middle Eastern ports. At the same time, in Europe and America, the production technology of low-sulfur marine fuels can change significantly, taking into account the characteristics of the extracted oils. Therefore, European oils, the characteristics of which are shown in Figure 17 and in Table 9, have a significantly lower sulfur content, so their residual fractions can be used in VLSFO compositions in significant quantities without prior treatment.

Taking into account the properties of Western European oils presented above, as well as their relative production volume, it is possible to calculate the approximate characteristics of "average" fuel oil: sulfur content–0.80%; amount of metals: vanadium–10.7 ppm, nickel–4.7 ppm, sum–15.4 ppm. Consequently, the properties of "average" straight-run fuel oil from Western Europe differ significantly from the quality parameters of Middle Eastern fuel oil and allow it to be used as the basis of low-sulfur marine fuels. These qualitative characteristics of AR will be used later in Section 3.4 in the study of batches of Western European ports [76]. In North America, the properties of extracted oil vary significantly more widely (Figure 18). In addition, the United States, being one of the largest importers of oil, further expands the permissible range of characteristics of oils and, as a result, AR by using both high-sulfur Middle Eastern oils and extremely low-sulfur oils of Western Europe. This variety makes it difficult to accurately determine the most probable properties of AR, which, together with the total small number of fuel samples taken for analysis from North America, makes it impossible to conduct a full-fledged analysis of the composition of the low-sulfur residual component, therefore, in further calculations, the properties of fractions similar to Singapore and the Middle East will be used.

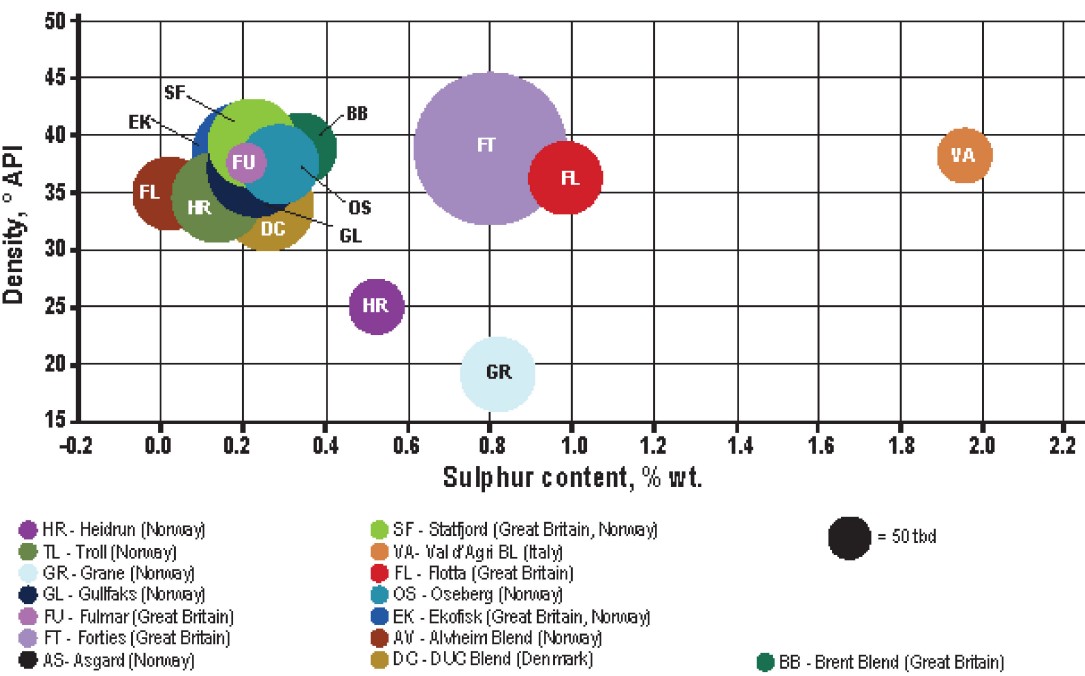

**Figure 17.** Quantity and quality of Western European oil [74].

**Table 9.** The content of metals and sulfur in various grades of commercial oils of Western Europe [75].

| Impurity | The Content of Metals and Sulfur in Grades of Commercial Oil, mg/kg and % wt. | | | | | | | | |
|---|---|---|---|---|---|---|---|---|---|
| | Forties | Troll | Brent | Aasgard | Ekofisk | Grane | Gullfaks | Oseberg | Statfjord |
| Vanadium | 9.2 | 0.9 | 6.3 | 0.8 | 1.9 | 9.6 | 0.8 | 1.0 | 1.7 |
| Nickel | 3.3 | 0.8 | 1.2 | 0.1 | 3.0 | 3.1 | 0.7 | 1.0 | 1.1 |
| Total metals | 12.5 | 1.7 | 7.5 | 0.9 | 4.9 | 12.7 | 1.5 | 2.0 | 2.8 |
| Sulfur | 0.82 | 0.17 | 0.35 | 0.11 | 0.21 | 0.62 | 0.18 | 0.2 | 0.2 |

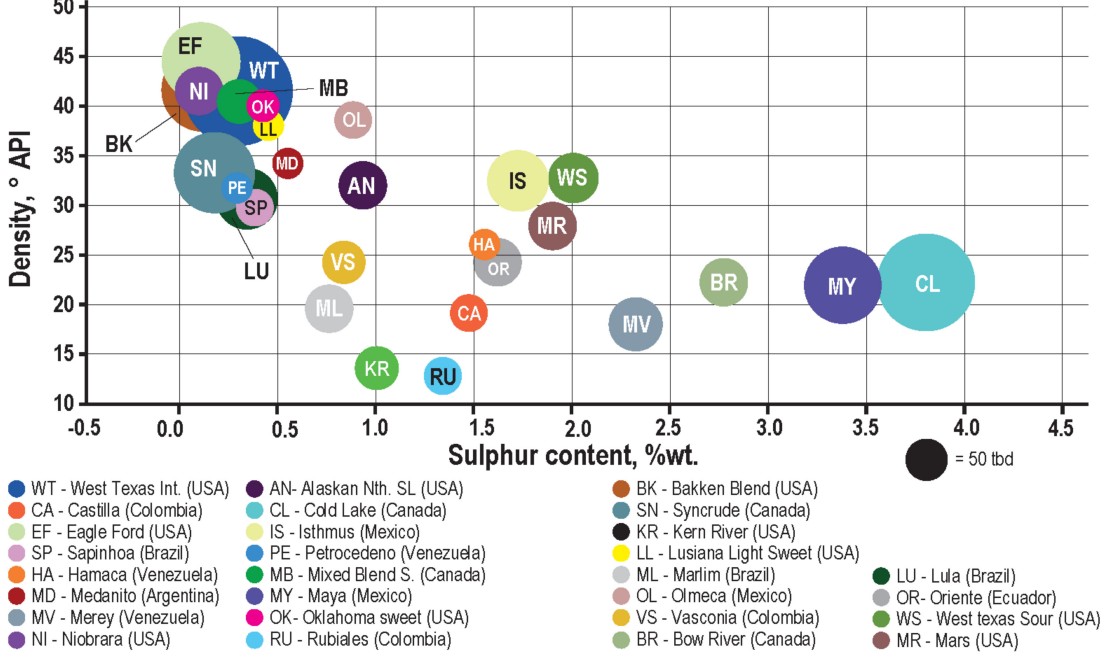

**Figure 18.** Quantity and quality of American oil [77].

*3.3. Analysis of VLSFO Fuel Compositions of RMG 380 Brand (Other Ports)*

It is logical to continue further analysis of fuel compositions of low-sulfur marine fuels from ports with a location similar to Singapore and the initial oil flows: Asian (Hong Kong) and Middle Eastern (Dubai, Fujairah, Jebel Ali Khor Fakkan), and then study North American (Corpus Christi, Pascagoula, GOLA terminal) and European ports (Amsterdam, Rotterdam, Antwerp). Since there are significantly fewer batches in these ports in the Lloyd's Register monitoring data, the study of fuel compositions will be carried out according to a simplified (graphical) methodology and for all fuel suppliers at once [78].

3.3.1. Hong Kong Port

During the period under review, 137 bunkering operations with low-sulfur marine fuel were carried out in this port. Graphically, the viscosity–density properties of these batches are presented in Figure 19.

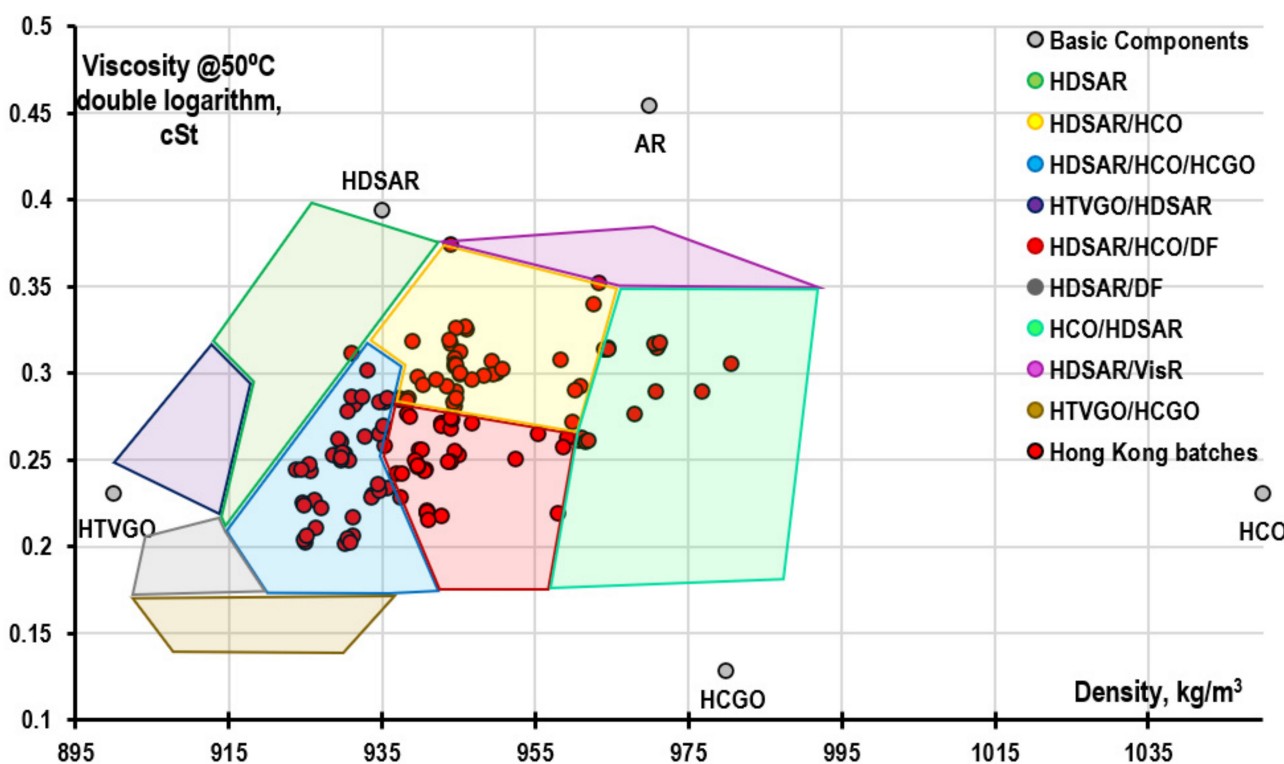

**Figure 19.** Hong Kong Port parties in logarithmic coordinates [79].

All the Hong Kong port batches taken for the study correspond to the existing typing structure for various fuel compositions. The results of the distribution are presented in Table 10.

The most popular fuel compositions of low-sulfur marine fuels in the port of Hong Kong are extremely similar to the compositions of the port of Singapore: the key components are HDSAR and HCO [81]. Moreover, the share of the HDSAR is slightly lower than in Singapore (62% vs. 65%), and the share of HCO, on the contrary, is higher (20% vs. 18%). The share of the remaining components remained almost unchanged with the exception of vacuum gas oil (3% compared to 5%), compositions based on which were practically not released in Hong Kong.

**Table 10.** Distribution of shipments in the port of Hong Kong by type [80].

| Type No. | Approximate Composition of the Composition, % wt. | | | | | | Type Code Name | Number of Batches |
|---|---|---|---|---|---|---|---|---|
| | HDSAR | AR /VR | HCO | HCGO | HTVGO | HTDF | | |
| 1 | 80–95 | 0–10 | 0–7 | <3 | 0–20 | 0–15 | HDSAR | 1 |
| 2 | 60–75 | <3 | 15–35 | 0–5 | 0–10 | 0–8 | HDSAR + HCO | 39 |
| 3 | 55–80 | <3 | 0–20 | 0–15 | <3 | 5–20 | HDSAR + HCO + HCGO | 52 |
| 4 | 30–60 | | <3 | 0–10 | 35–60 | 5–15 | HTVGO + HDSAR | 0 |
| 5 | 40–70 | <3 | 10–35 | 0–12 | 0–7 | 10–15 | HDSAR + HCO + DF | 33 |
| 6 | 60–75 | <3 | 0–10 | | <3 | 15–30 | HDSAR + DF | 0 |
| 7 | 35–55 | <3 | 35–50 | 0–5 | 0–10 | 0–15 | HCO + HDSAR | 12 |
| 8 | 60–75 | 0–10 | 0–25 | <3 | 0–10 | <3 | HDSAR + VR | 0 |
| 9 | 0–7 | | <3 | 15–30 | 50–70 | 0–10 | HTVGO + HCGO | 0 |

### 3.3.2. Ports of the Middle East

During the period under review, 680 bunkering operations with low-sulfur marine fuel were carried out in the ports of the Middle East [82]. Graphically, the viscosity–density properties of these batches are shown in Figure 20.

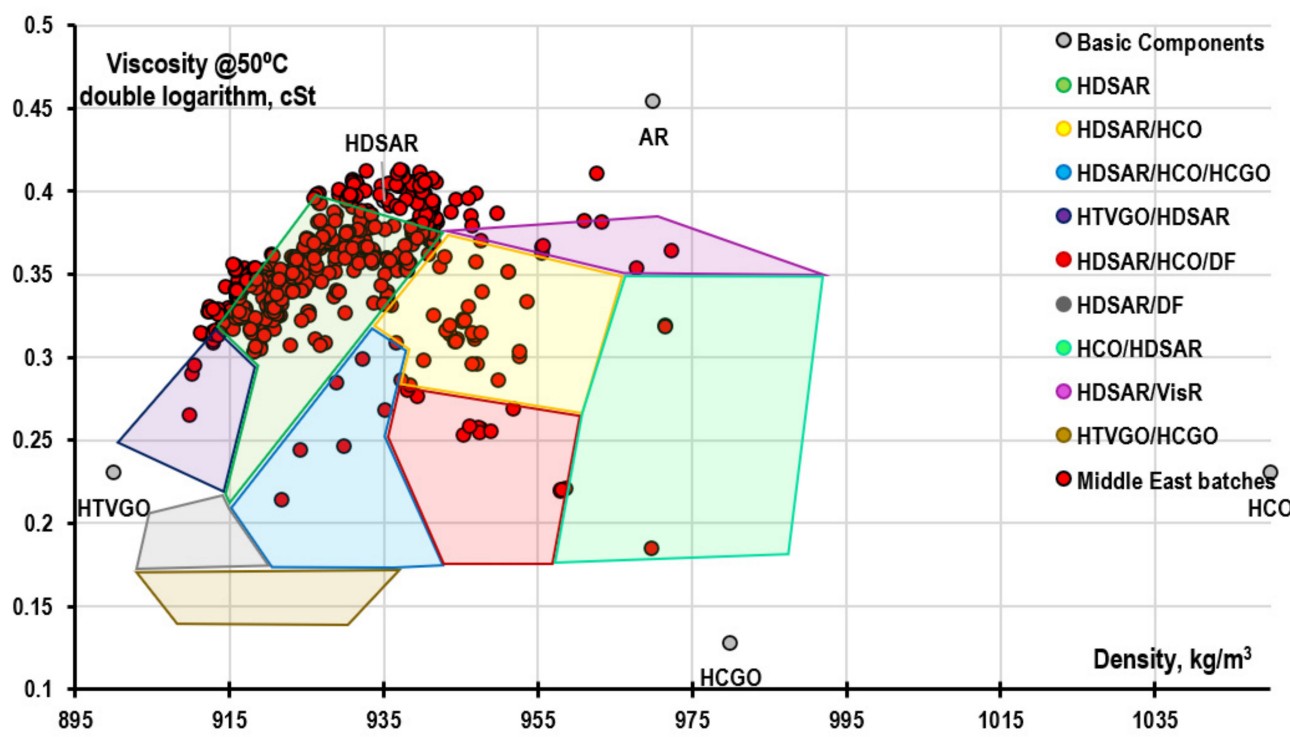

**Figure 20.** Lots of ports in the Middle East in logarithmic coordinates [83].

As is easy to see from Figure 13, the batches of marine fuel from Middle Eastern ports are extremely similar to each other in terms of viscosity–density characteristics and are concentrated in the area characteristic of HDSAR. According to their location, it is possible with a fairly high degree of probability to declare the most popular composition of marine fuels in these ports–type 1 "HDSAR". Moreover, the concentration of diluents (diesel fuels, vacuum gas oil) and non-straight-run heavy gas oils in the case of the presented batches is extremely small. Some samples of marine fuels may shift relative to the selected properties of hydrotreated fuel oil (the base component) to a denser or more viscous zone, however, it is obvious that this is precisely due to the characteristics of the HDSAR, which, depending on the operating mode and the raw materials used, may vary slightly, and not the presence of additional mixed components.

The dominant position of hydrotreated fuel oil in the composition can also be confirmed by analyzing the capacities available in the Middle East for refining atmospheric residues. For example, the Al-Zur oil refinery, located in Kuwait, has the world's largest AR hydrotreating plant, with a capacity of 11 million tons calculated per the target product [84]. In the presence of such large resources of the base low-sulfur component, it is not surprising that it is completely dominant in existing fuel compositions. The results of the distribution of Middle Eastern batches of low-sulfur marine fuel are presented in Table 11.

**Table 11.** Distribution of shipments of ports of the Middle East by type [85].

| Type No. | Approximate Composition of the Composition, % wt. | | | | | | Type Code Name | Number of Batches |
|---|---|---|---|---|---|---|---|---|
| | HDSAR | AR /VR | HCO | HCGO | HTVGO | HTDF | | |
| 1 | 80–95 | 0–10 | 0–7 | <3 | 0–20 | 0–15 | HDSAR | 620 |
| 2 | 60–75 | <3 | 15–35 | 0–5 | 0–10 | 0–8 | HDSAR + HCO | 33 |
| 3 | 55–80 | <3 | 0–20 | 0–15 | <3 | 5–20 | HDSAR + HCO + HCGO | 8 |
| 4 | 30–60 | | <3 | 0–10 | 35–60 | 5–15 | HTVGO + HDSAR | 3 |
| 5 | 40–70 | <3 | 10–35 | 0–12 | 0–7 | 10–15 | HDSAR + HCO + DF | 11 |
| 6 | 60–75 | <3 | 0–10 | | <3 | 15–30 | HDSAR + DF | 0 |
| 7 | 35–55 | <3 | 35–50 | 0–5 | 0–10 | 0–15 | HCO + HDSAR | 5 |
| 8 | 60–75 | 0–10 | 0–25 | <3 | 0–10 | <3 | HDSAR + VR | 0 |
| 9 | 0–7 | | <3 | 15–30 | 50–70 | 0–10 | HTVGO + HCGO | 0 |

The dominant position of HDSAR affects the popularity of type 1 "HDSAR" compositions, as well as its total content in the amount of marine fuels–87% [86]. The remaining components are presented in much smaller quantities, and the greatest relative decrease is noticeable in HCO, which is used in Middle Eastern compositions in an average amount of 3.6%, which is more than 5 times lower than in Singapore. The content of diesel fractions also fell from 8% to 4%, and the proportion of vacuum gas oil, on the contrary, practically did not decrease. Nevertheless, it is worth noting that, given the well-known technology for producing most of the low-sulfur fuels in the ports under study, vacuum gas oil may not be directly part of the composition, but may be a parameter of different operating modes of the fuel oil hydrotreating unit. That is, in the mode of maximum distillate extraction, a heavier fuel is obtained, which in this work is classified as practically pure fuel oil, and in the mode of smaller extraction, a relatively low-viscosity fuel is obtained, which in the process of composition modeling is interpreted as AR+ vacuum gas oil.

### 3.3.3. Ports of North America

During the period under review, 54 bunkering operations with low-sulfur marine fuel were carried out in the ports of North America [87]. Graphically, the viscosity–density properties of these batches are shown in Figure 21.

In the ports of North America, there is a fairly large variety of compositions of marine fuels. There is also practically pure HDSAR (in close proximity to "HDSAR point"), and there are mixtures of it with HCO both in small ("HDSAR + HCO") and in very large quantities ("HCO + HDSAR"). It is also possible to notice a generally lower location of the points in Figure 21 within the limits of the density of 950–980 kg/m$^3$, which allows us to assume a higher proportion of heavy coking gas oil in mixtures of these compositions than similar types of fuels in Singapore. The results of the distribution of North American batches of low-sulfur fuel are presented in Table 12.

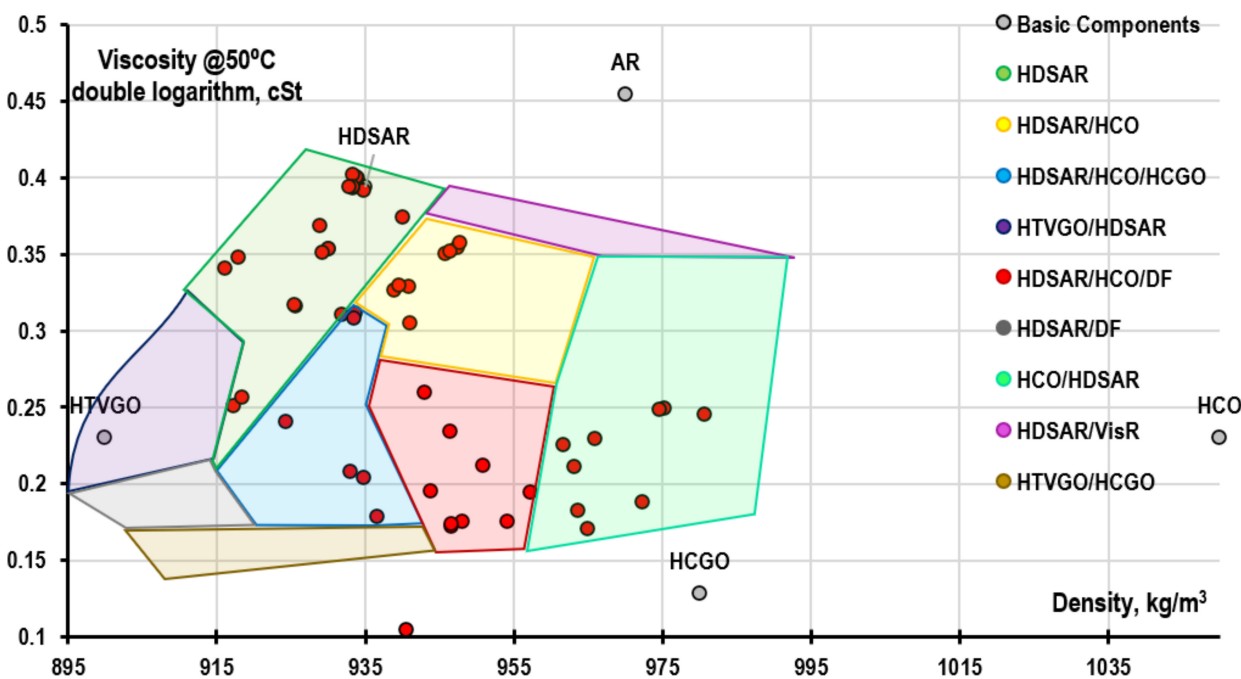

**Figure 21.** Lots of North American ports in logarithmic coordinates [87].

**Table 12.** Distribution of shipments of North American ports by type [88].

| Type No. | Approximate Composition of the Composition, wt. % | | | | | | Type Code Name | Number of Batches |
| --- | --- | --- | --- | --- | --- | --- | --- | --- |
| | HDSAR | AR/VR | HCO | HCGO | HTVGO | HTDF | | |
| 1 | 80–95 | 0–10 | 0–7 | <3 | 0–20 | 0–15 | HDSAR | 22 |
| 2 | 60–75 | <3 | 15–35 | 0–5 | 0–10 | 0–8 | HDSAR + HCO | 8 |
| 3 | 55–80 | <3 | 0–20 | 0–15 | <3 | 5–20 | HDSAR + HCO + HCGO | 6 |
| 4 | 30–60 | | <3 | 0–10 | 35–60 | 5–15 | HTVGO + HDSAR | 0 |
| 5 | 40–70 | <3 | 10–35 | 0–12 | 0–7 | 10–15 | HDSAR + HCO + DF | 9 |
| 6 | 60–75 | <3 | 0–10 | | <3 | 15–30 | HDSAR + DF | 0 |
| 7 | 35–55 | <3 | 35–50 | 0–5 | 0–10 | 0–15 | HCO + HDSAR | 9 |
| 8 | 60–75 | 0–10 | 0–25 | <3 | 0–10 | <3 | HDSAR + VR | 0 |

Due to the rather large diversity of compositions, the total share of HDSAR in batches is about 70%. The average content of HCO is quite high (12%), but it is inferior to the indicators of Singapore and Hong Kong, which is compensated by an increased proportion of heavy coking gas oil (7.6% versus 4–5% in Asian ports).

3.3.4. Ports of Western Europe

Two hundred and sixteen bunkering operations with low-sulfur marine fuel were carried out in the ports of Western Europe during the period under review. Graphically, the viscosity–density properties of these batches are shown in Figure 22.

In European ports, all significantly higher viscosity values are observed, which can be explained by the additional involvement of low-sulfur straight-run high-viscosity fractions (AR) or visbreaking residues, the resources of which in Europe are significantly larger due to the specifics of the oils produced here. For the same reason, a separate modeling of the composition of samples was carried out for Western European batches based on the following list of components:

- low-sulfur straight-run AR—the sulfur content of 0.8% was calculated earlier based on the properties of the extracted oils; density 970 kg/m$^3$; viscosity at 50 °C 700 cSt.

- Hydrotreated/hydrodesulfurized atmospheric residue (HDSAR) is a standard product of a hydrotreating plant: sulfur content 0.3%; density 930 kg/m$^3$; viscosity at 50 °C 300 cSt.
- Visbreaking residue (VisR) is obtained from low-sulfur tar and has a sulfur content of 1.2%; density of 1050 kg/m$^3$; viscosity at 50 °C 3000 cSt.
- Heavy catalytic cracker cycle oil (HCO) has properties similar to those previously adopted for ports in Asia, the Middle East, and America: sulfur content 0.5%, density 1050 kg/m$^3$, and viscosity at 50 °C 50 cSt.
- Hydrotreated vacuum gas oil (HTVGO) is a standard product of a catalytic cracking feedstock preparation unit: 0.1% sulfur content, density 900 kg/m$^3$, viscosity at 50 °C 50 cSt.
- Straight-run vacuum gas oil (VGO) is a low-sulfur vacuum gas oil obtained from Western European oils: sulfur content 0.6%, density 910 kg/m$^3$, viscosity at 50 °C 70 cSt.
- Diesel fuel (DF)—depending on the sulfur content in marine fuel, diesel fuel from straight-run to ultra-low-sulfur can be used. The properties are within the following limits: sulfur content 0.001–1.0%, density 828–840 kg/m$^3$, viscosity at 50 °C 2.0–2.4 cSt.

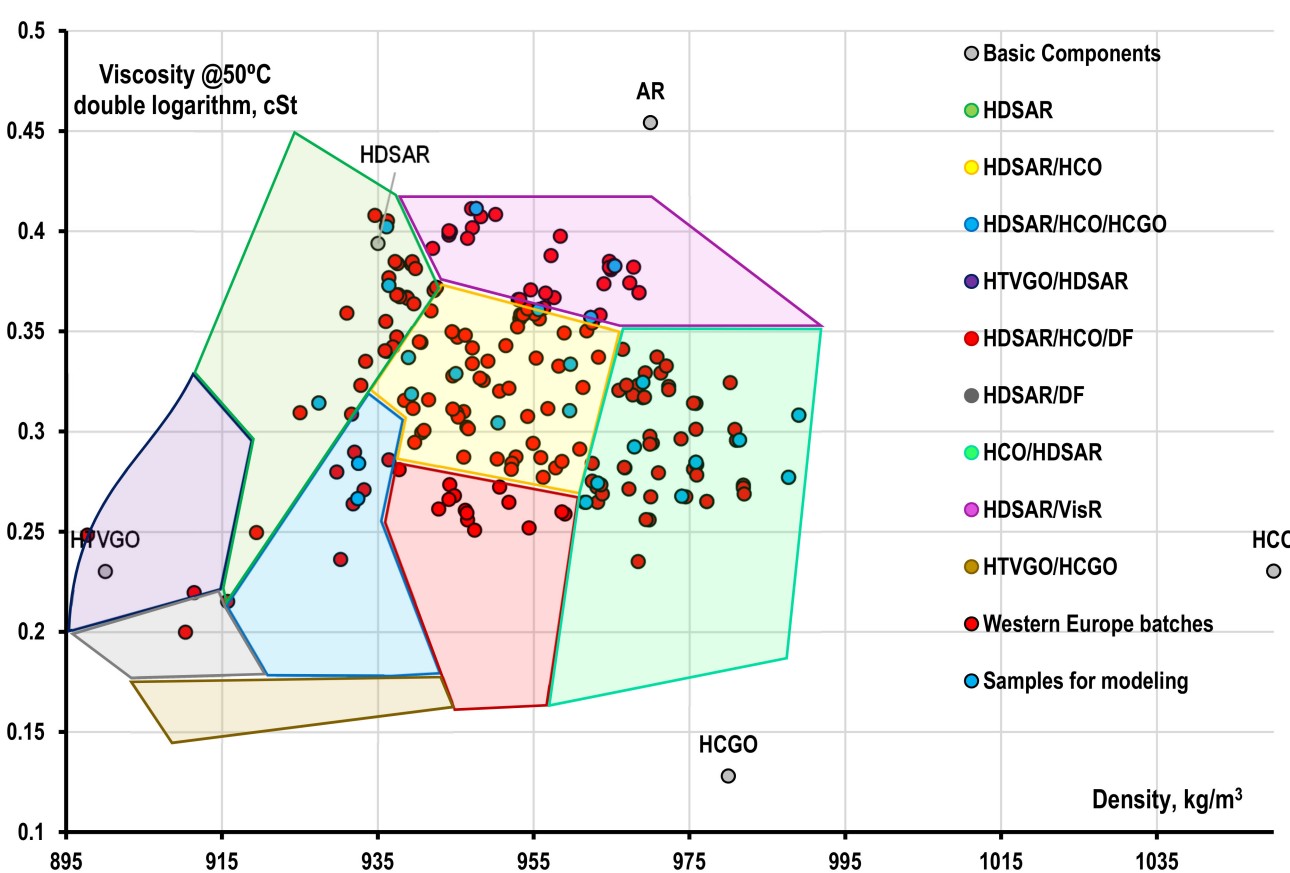

**Figure 22.** Lots of Western European ports in logarithmic coordinates (standard typing) [89].

In order to reduce the amount of work, 24 samples were selected for modeling, distributed over the entire viscosity–density characteristic of European batches. The results of modeling their composition are presented in Table 13.

**Table 13.** Selected samples of VLSFO ports of Western Europe and the results of modeling their composition [90,91].

| Sample No. | Density 15 °C, kg/m³ | Kinematic Viscosity at 50 °C, cSt | Sulfur Content, % | Vanadium Content, ppm | Aluminum and Silicon Content, ppm | Nickel Content, ppm | Calculated Content of Various Components in Fuel Samples | | | | | |
|---|---|---|---|---|---|---|---|---|---|---|---|---|
| | | | | | | | HDSAR | HTVGO | AR | VGO | HCO | DF |
| 1 | 947.6 | 378.8 | 0.50 | 19 | 12 | 15 | 55 | 2 | 41 | 0 | 2 | 0 |
| 2 | 936.1 | 335 | 0.50 | 7 | 4 | 10 | 56 | 0 | 35 | 9 | 0 | 0 |
| 3 | 965.4 | 259.2 | 0.49 | 7 | 10 | 15 | 44 | 1 | 35 | 0 | 20 | 0 |
| 4 | 936.4 | 229.4 | 0.49 | 13 | 9 | 13 | 16 | 31 | 53 | 0 | 0 | 0 |
| 5 | 955.6 | 197.5 | 0.50 | 12 | 26 | 22 | 12 | 25 | 50 | 0 | 13 | 0 |
| 6 | 962.3 | 188.4 | 0.48 | 4 | 29 | 12 | 21 | 18 | 41 | 0 | 20 | 0 |
| 7 | 938.9 | 148.9 | 0.50 | 9 | 19 | 12 | 0 | 36 | 60 | 0 | 1 | 4 |
| 8 | 959.7 | 143.3 | 0.50 | 8 | 25 | 12 | 0 | 28 | 51 | 0 | 19 | 2 |
| 9 | 945 | 135.7 | 0.48 | 8 | 10 | 4 | 0 | 35 | 54 | 0 | 8 | 3 |
| 10 | 969 | 129.3 | 0.49 | 11 | 18 | 13 | 0 | 23 | 45 | 0 | 29 | 3 |
| 11 | 939.3 | 121.1 | 0.49 | 7 | 14 | 10 | 0 | 34 | 57 | 0 | 4 | 5 |
| 12 | 927.4 | 115.4 | 0.49 | 4 | 18 | 3 | 0 | 33 | 38 | 28 | 0 | 1 |
| 13 | 959.6 | 110.6 | 0.49 | 8 | 11 | 12 | 0 | 25 | 49 | 0 | 21 | 4 |
| 14 | 989 | 108 | 0.48 | 0 | 45 | 6 | 0 | 18 | 36 | 0 | 45 | 1 |
| 15 | 950.4 | 103.6 | 0.48 | 8 | 9 | 11 | 0 | 29 | 51 | 0 | 15 | 6 |
| 16 | 981.4 | 94.7 | 0.50 | 9 | 26 | 8 | 0 | 14 | 42 | 0 | 40 | 4 |
| 17 | 967.9 | 91.3 | 0.47 | 14 | 31 | 15 | 0 | 23 | 42 | 0 | 31 | 5 |
| 18 | 975.8 | 84.4 | 0.49 | 6 | 24 | 1 | 0 | 16 | 42 | 0 | 37 | 6 |
| 19 | 932.5 | 83.9 | 0.49 | 6 | 7 | 9 | 0 | 29 | 60 | 0 | 1 | 11 |
| 20 | 987.7 | 78.2 | 0.50 | 4 | 42 | 8 | 0 | 9 | 39 | 0 | 46 | 6 |
| 21 | 963.2 | 76 | 0.48 | 14 | 17 | 11 | 0 | 19 | 45 | 0 | 28 | 8 |
| 22 | 974 | 71.2 | 0.50 | 1 | 24 | 2 | 0 | 11 | 44 | 0 | 37 | 9 |
| 23 | 932.4 | 70.4 | 0.50 | 3 | 47 | 8 | 0 | 24 | 60 | 0 | 3 | 14 |
| 24 | 961.7 | 69.1 | 0.50 | 15 | 24 | 10 | 0 | 14 | 48 | 0 | 27 | 11 |

As can be seen from the Table 13, in the ports of Western Europe, HDSAR was used only in a small number of compositions: six samples for modeling, which correspond to 70 real batches. Moreover, taking into account the properties of European oils, these compositions may also not include treated residual components, but be based not on "average" low-sulfur fuel oil (0.8% sulfur), but on a real sample with an even lower sulfur content and certain viscosity–density characteristics that deviate somewhat from the "standard" ones. Consequently, a decrease in the sulfur content down to 0.5% in low-sulfur straight-run AR used for modeling makes it possible to additionally abandon the use of HDSAR in samples 3, 5, and 6. The remaining samples cannot be obtained only by manipulating the sulfur content in the AR and certain viscosity–density properties of the residual component are needed for their modeling. This is especially evident for samples 1 and 2, in which the AR content is over 90%. In their case, it is not possible to significantly affect the properties of the composition by adding other fractions, since their proportion is too small.

Taking into account the aforementioned, it is proposed to count all fuel oils in the batches of Western Europe as straight-run, which will facilitate further division into principal types, which is presented in Table 14 and Figure 23.

**Table 14.** Distribution of shipments of Western European ports by type [92].

| Type No. | Approximate Composition of the Composition, % wt. | | | | Type Code Name | Number of Batches |
|---|---|---|---|---|---|---|
| | AR | HTVGO + VGO | HCO | DF | | |
| 1 | 90–100 | 0–10 | <3 | <3 | AR | 19 |
| 2 | 45–90 | 0–25 | 15–35 | 0–5 | AR + HCO | 42 |
| 3 | 50–70 | 20–40 | 0–15 | 0–15 | AR + VGO | 98 |
| 4 | 30–45 | 50–70 | <3 | 0–15 | VGO + AR | 7 |
| 5 | 40–50 | 0–25 | 15–35 | 5–15 | AR + HCO + DF | 29 |
| 6 | 30–45 | 0–20 | 35–50 | 0–10 | HCO + AR | 21 |

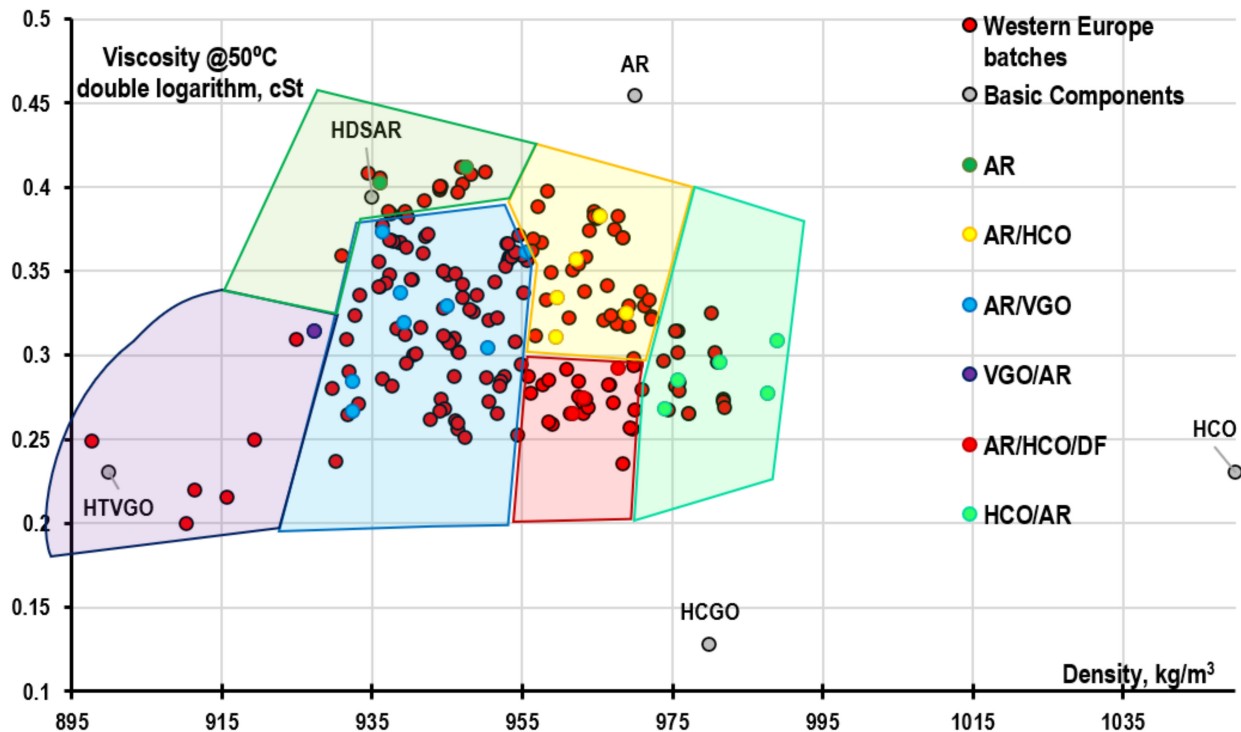

**Figure 23.** Division of all studied shipments of Western European ports by type [93].

By analogy with the batches of the port of Singapore for Western European samples, a classification was made according to the principal types of compositions depending on the composition of specific fuels and, accordingly, on their viscosity–density properties. The most popular composition was the fuel composition "AR + VGO", consisting mainly of two components of straight-run AR (50–70%) and hydrotreated vacuum gas oil (20–40%), in addition, the presence of up to 15% of HCO and diesel fuel is allowed in the composition. In addition to this composition, compositions that include a significant amount of heavy cracker cycle oil are popular in the ports of Western Europe: 71 samples with an HCO content of 15–35% and 21 samples with a content of 35–50%. Thus, European ports differ significantly from Middle Eastern and Asian ports in terms of the types of fuel compositions used. The main reason for this is the possibility of involving straight-run residual components in the VLSFO composition in quantities of more than 10–15% (approximate restrictions for "average" Middle Eastern AR). In the case of Western Europe, the average content of straight-run AR in the composition reaches 59%. Another distinctive feature of European batches is the increased proportion of vacuum gas oil in the compositions: 21% compared to 5% for Singapore, which is probably due to the need for a more intensive reduction in the sulfur content in the fuel compared to compositions based on HDSAR, for which it is not necessary to reduce the sulfur content. The content of HCO in the compositions is approximately at the level of global values: 15% vs. 11–20% in the world.

*3.4. Analysis of RME 180 VLSFO Fuel Compositions*

The fuel grade RME 180 (or RMG 180/RMF 180) sharply loses popularity to the higher viscosity RMG 380. In all the ports studied, only 227 bunkerings with this type of fuel were recorded, so it makes little sense to consider ports/bunkering zones separately due to the small amount of data and the inability to draw qualitative conclusions based on them. Instead, the key features of fuel distribution by the principal types identified earlier are considered. Graphically, the viscosity–density properties of RME 180 batches are shown in Figure 24.

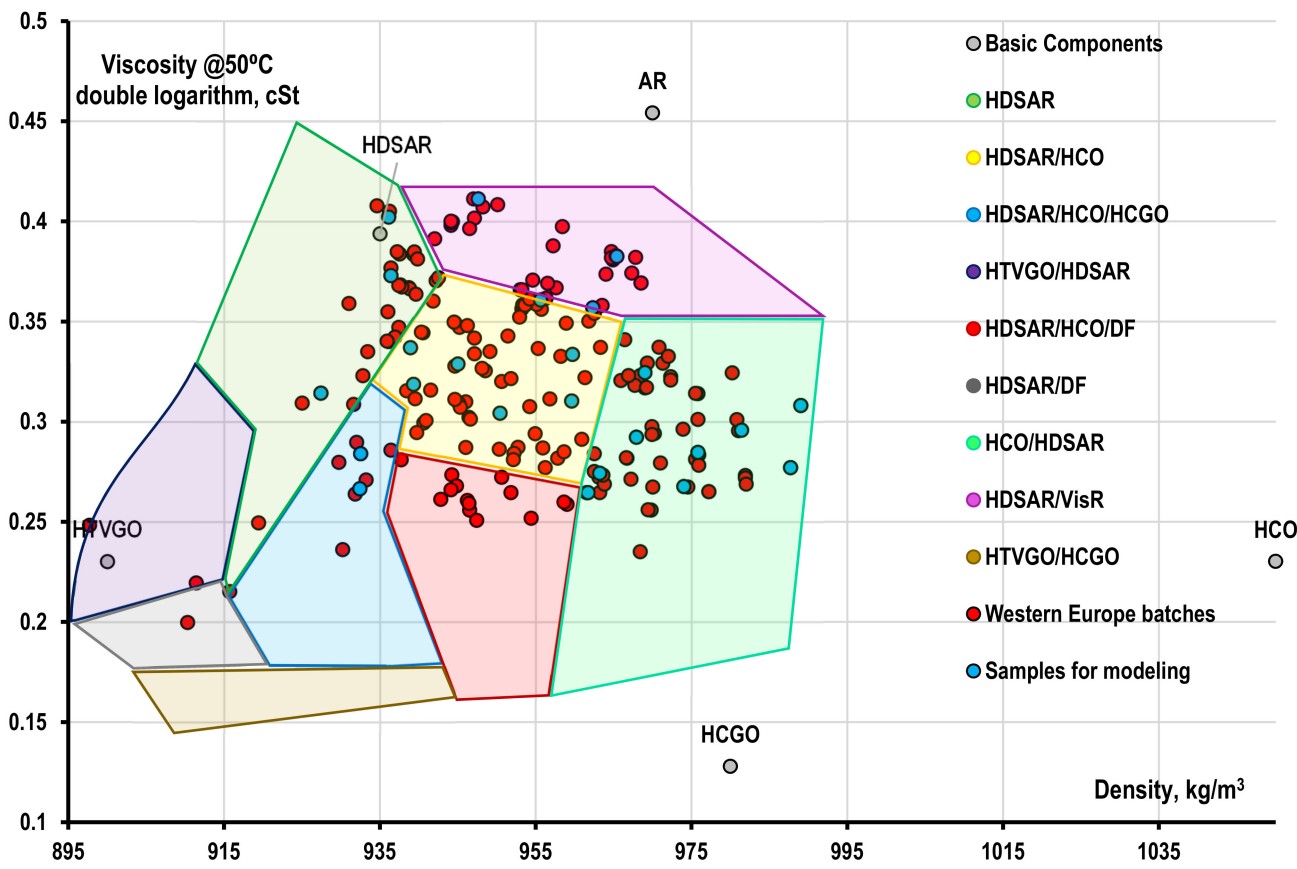

**Figure 24.** RME 180 batches in logarithmic coordinates [94].

Figure 24 shows that the dominant type of composition is "HDSAR", which is quite simply explained if we take into account the distribution of RME 180 batches by bunkering zones: 174 deliveries (76%) were made from ports in the Middle East. A strong imbalance towards Middle Eastern supplies has made the most popular fuel composition "HDSAR" in this region popular all over the world. If we exclude the parties of the Middle East from the analysis and consider the rest of the world, then we can notice an approximately uniform distribution of batches between types 1–3 and 5, which generally corresponds to the distribution of batches of the RMG 380 brand. This conclusion is quite obvious if we take into account the previously stated distribution of batches of the RMG 380 brand by viscosity–approximately 85–90% have a viscosity below 180 cSt, that is, they formally fall under the RMG 180 brand (or RME 180/RMF 180 if all other requirements are met). Thus, in principle, the fuel compositions of these brands do not differ from each other, with the exception of some types of fuels that go beyond the viscosity limits of the RME 180 brand–types "HDSAR +VR" and "HDSAR" with a minimum content of diluents (practically pure HDSAR). The results of the distribution of Middle Eastern and other world batches of low-sulfur fuel RME 180 are presented in Tables 15 and 16.

In the Middle Eastern batches of RME 180, the same components are used as in the samples of the RMG 380 brand, in almost unchanged concentrations: HDSAR—86%, HCO–4%, vacuum gas oil–6%, diesel fractions–4%. In the rest of the world, on the contrary, for the RME 180 brand, compared with RMG 380, there is a significant change in the average content of HDSAR and HCO–a decrease from 62–65% to 60% and an increase from 18–22% to 25%, respectively. The share of batches from Western European ports in the total number of samples of the RME 180 brand is small (less than 10%), so it makes little sense to consider them separately, taking into account the specific component composition.

**Table 15.** Distribution of shipments of RME 180 from ports of the Middle East by type [95].

| Type No. | Approximate Composition of the Composition, % wt. | | | | | | Type Code Name | Number of Batches |
|---|---|---|---|---|---|---|---|---|
| | HDSAR | AR /VR | HCO | HCGO | HTVGO | HTDF | | |
| 1 | 80–95 | 0–10 | 0–7 | <3 | 0–20 | 0–15 | HDSAR | 151 |
| 2 | 60–75 | <3 | 15–35 | 0–5 | 0–10 | 0–8 | HDSAR + HCO | 15 |
| 3 | 55–80 | <3 | 0–20 | 0–15 | <3 | 5–20 | HDSAR + HCO + HCGO | 4 |
| 4 | 30–60 | | <3 | 0–10 | 35–60 | 5–15 | HTVGO + HDSAR | 3 |
| 5 | 40–70 | <3 | 10–35 | 0–12 | 0–7 | 10–15 | HDSAR + HCO + DF | 0 |
| 6 | 60–75 | <3 | 0–10 | | <3 | 15–30 | HDSAR + DF | 0 |
| 7 | 35–55 | <3 | 35–50 | 0–5 | 0–10 | 0–15 | HCO + HDSAR | 1 |
| 8 | 60–75 | 0–20 | 0–25 | <3 | 0–10 | <3 | HDSAR + BO | 0 |
| 9 | 0–7 | | <3 | 15–30 | 50–70 | 0–10 | HTVGO + HDSAR | 0 |

**Table 16.** Distribution of shipments of RME 180 from world ports by type (excluding the Middle East) [96].

| Type No. | Approximate Composition of the Composition, % wt. | | | | | | Type Code Name | Number of Batches |
|---|---|---|---|---|---|---|---|---|
| | HDSAR | AR /VR | HCO | HCGO | HTVGO | HTDF | | |
| 1 | 80–95 | 0–10 | 0–7 | <3 | 0–20 | 0–15 | HDSAR | 7 |
| 2 | 60–75 | <3 | 15–35 | 0–5 | 0–10 | 0–8 | HDSAR + HCO | 20 |
| 3 | 55–80 | <3 | 0–20 | 0–15 | <3 | 5–20 | HDSAR + HCO + HCGO | 7 |
| 4 | 30–60 | | <3 | 0–10 | 35–60 | 5–15 | HTVGO+ HDSAR | 1 |
| 5 | 40–70 | <3 | 10–35 | 0–12 | 0–7 | 10–15 | HDSAR + HCO + DF | 0 |
| 6 | 60–75 | <3 | 0–10 | | <3 | 15–30 | HDSAR + DF | 0 |
| 7 | 35–55 | <3 | 35–50 | 0–5 | 0–10 | 0–15 | HCO + HDSAR | 18 |
| 8 | 60–75 | 0–20 | 0–25 | <3 | 0–10 | <3 | HDSAR + VR | 0 |
| 9 | 0–7 | | <3 | 15–30 | 50–70 | 0–10 | HTVGO + HDSAR | 0 |

*3.5. Analysis of VLSFO Fuel Compositions of RMD 80 Brand*

RMD 80 fuels are even less popular in the world market than RME 180. During the study period, according to Lloyd's Register, there were only 49 bunkers with this type of fuel. At the same time, the majority (about 50%) was carried out in the ports of the Middle East, another 30% in American ports, about 15% in Singapore, and only 5% in Europe. For the RMD80 brand, as well as to the RME 180, the statement about incomplete labeling of all batches suitable for viscosity and their more frequent use in the form of RMG 380 is applicable. Thus, about 40% of RMG 380 fuel samples have a viscosity below 80 cSt, therefore, the method of coding fuel compositions of this brand can be fully used for less viscous RMD 80. Graphically, the viscosity–density properties of RMD 80 batches are shown in Figure 25.

Figure 25 shows that the approximate distribution of batches by principal fuel compositions is similar to RMG 380, but there is a fairly obvious shift to the area of less viscous compositions "HTVGO + HDSAR", "HDSAR + DF", and "HTVGO + HCGO". The results of the distribution of the world batches of low-sulfur fuel RMD80 are presented in Table 17.

Forced mixing for viscosity characteristics compliance caused a decrease in the average proportion of HDSAR in the compositions to 55% and an increase in the content of diluents: hydrotreated vacuum gas oil–14%, diesel fractions–11%.

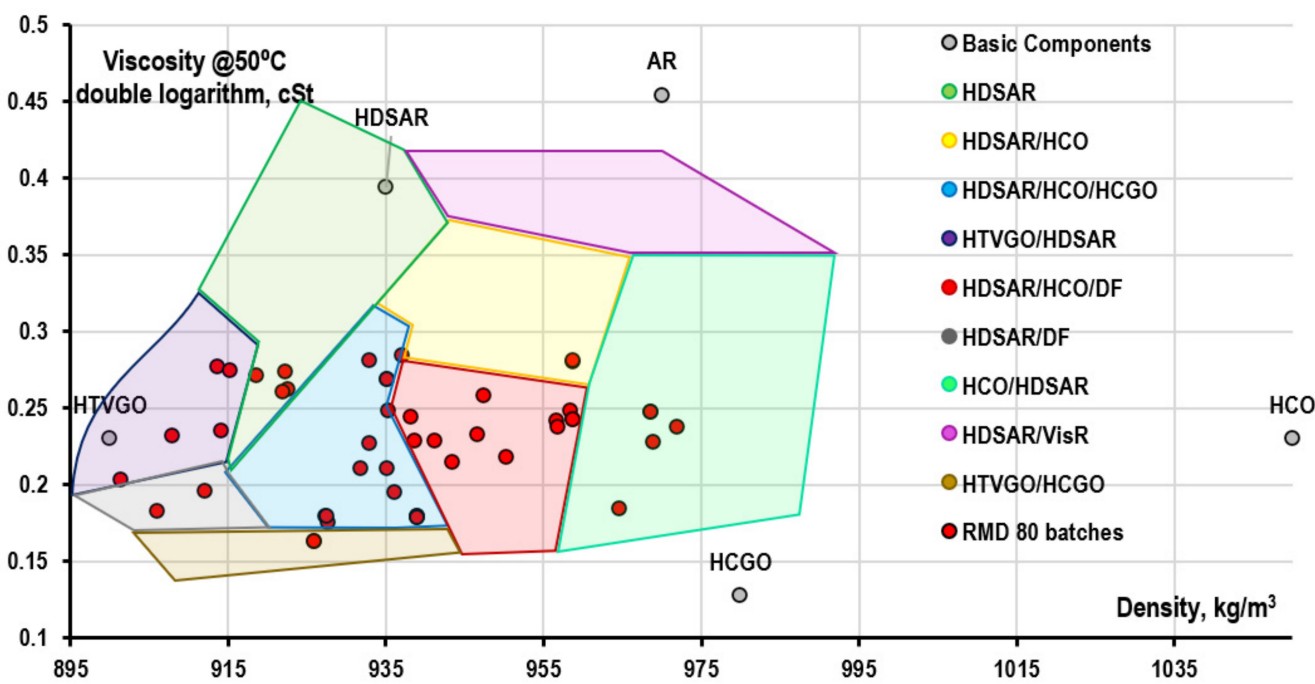

**Figure 25.** RMD 80 batches in logarithmic coordinates [97].

**Table 17.** Distribution of RMD batches of 80 world ports by type [98].

| Type No. | Approximate Composition of the Composition, % wt. | | | | | | Type Code Name | Number of Batches |
|---|---|---|---|---|---|---|---|---|
| | HDSAR | AR /VR | HCO | HCGO | HTVGO | HTDF | | |
| 1 | 80–95 | 0–10 | 0–7 | <3 | 0–20 | 0–15 | HDSAR | 5 |
| 2 | 60–75 | <3 | 15–35 | 0–5 | 0–10 | 0–8 | HDSAR + HCO | 2 |
| 3 | 55–80 | <3 | 0–20 | 0–15 | <3 | 5–20 | HDSAR + HCO + HCGO | 15 |
| 4 | 30–60 | | <3 | 0–10 | 35–60 | 5–15 | HTVGO + HDSAR | 5 |
| 5 | 40–70 | <3 | 10–35 | 0–12 | 0–7 | 10–15 | HDSAR + HCO + DF | 13 |
| 6 | 60–75 | <3 | 0–10 | | <3 | 15–30 | HDSAR + DF | 3 |
| 7 | 35–55 | <3 | 35–50 | 0–5 | 0–10 | 0–15 | HCO + HDSAR | 5 |
| 8 | 60–75 | 0–20 | 0–25 | <3 | 0–10 | <3 | HDSAR + VR | 0 |
| 9 | 0–7 | | <3 | 15–30 | 50–70 | 0–10 | HTVGO + HCGO | 1 |

### 3.6. Analysis of ULSFO Fuel Compositions

The basic fuel compositions of ultra-low-sulfur fuel are much simpler and more expectable, which can be explained by a smaller number of potential blending components: hydrotreated vacuum gas oil, hydrotreated diesel fuel, and light catalytic cracker cycle oil. Light cracker cycle oil (LCO) can be included in the composition of ULSFO either after hydrotreatment, during which it becomes indistinguishable in properties from diesel fuel, or in extremely small quantities that do not have a significant effect on the final properties of the compositions. As a result, all batches of ultra-low-sulfur marine fuels are concentrated at four points (Figure 26), the approximate component composition of which is described in Table 18.

The batches enclosed in the orange zone, despite the seemingly large viscosity interval of 30–90 cSt, with a very high probability consist of only one component–hydrotreated vacuum gas oil—but with slightly different boiling intervals.

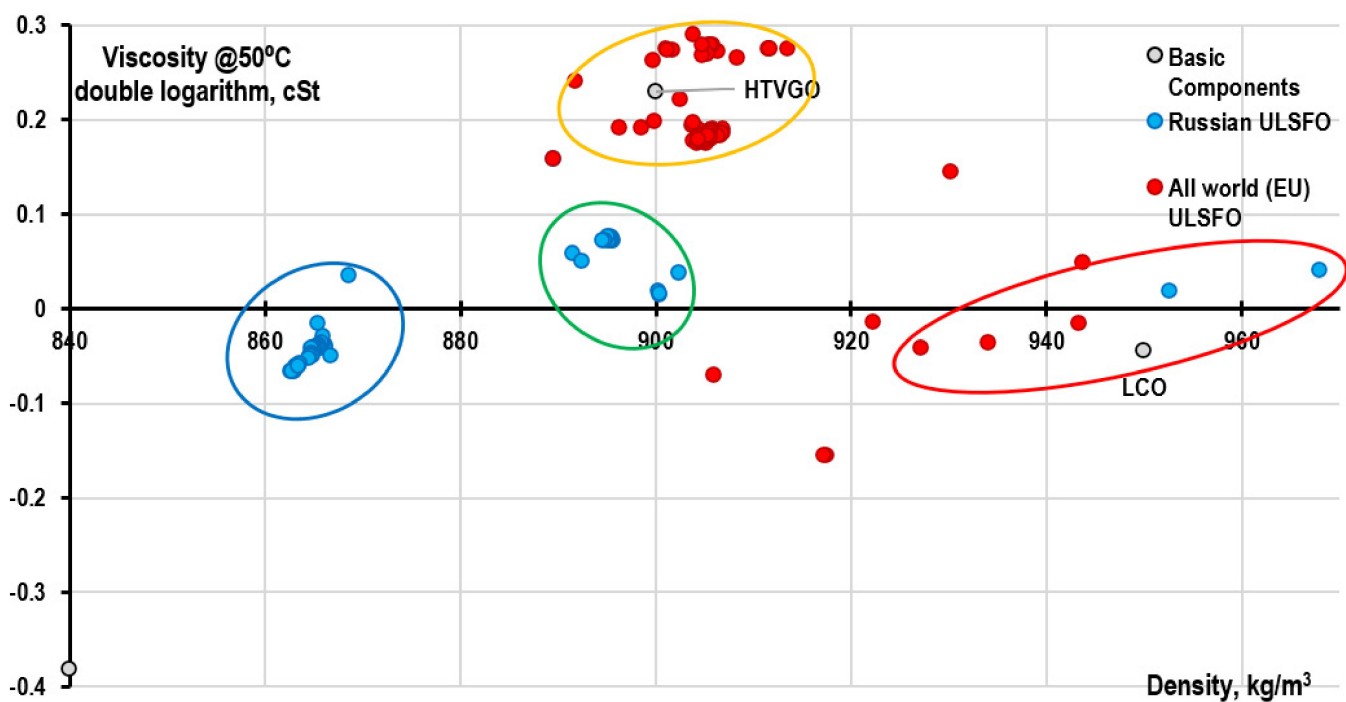

**Figure 26.** ULSFO batches in logarithmic coordinates [97].

**Table 18.** Composition of ULSFO fuel compositions [98].

| Color of the Selected Area | Density at 15 °C, kg/m³ | Kinematic Viscosity at 50 °C, cSt | Calculated Content of Various Components in Fuel Samples | | |
|---|---|---|---|---|---|
| | | | HTVGO | HTDF | LCO |
| Red | 950–970 | 10–15 | 0–10 | 0–10 | 85–100 |
| Orange | 890–920 | 30–90 | 90–100 | 0–5 | 0–5 |
| Green | 880–905 | 10–30 | 70–90 | 0–15 | 0–15 |
| Blue | 860–880 | 5–13 | 45–65 | 35–55 | 0 |

In addition to a fairly simple composition, ULSFO batches have another feature related to their distribution in bunkering zones, which strictly correlates with emission control zones, where, in fact, this fuel is required to be used. Outside of Western Europe and Russian Baltic ports, ULSFO is practically not implemented.

## 4. Conclusions

Despite the fact that most bunkered fuels belong to the RMG 380 brand, about half of the total volume of VLSFO has a kinematic viscosity in the range of 50–110 cSt, which indicates the frequent unification of labeling of various batches of marine fuels without reference to their specific properties. Furthermore, the peak distribution of the studied fuel batches by density characteristics is located in the range of 930–950 kg/m³, which allows us to make an assumption about the most popular component of VLSFO–hydrotreated fuel oil (density 920–940 kg/m³). At the same time, the viscosity of the average VLSFO sample is much less than the typical value for refined fuel oil–50–110 cSt versus 200–400 cSt. As the viscosity of fuels increases, a regular increase in the content of hydrotreated fuel oil occurs in them, which can be traced by the indicator of the content of vanadium and nickel.

The number of bunkers produced by fuel with ultra-low-sulfur content ULSD is noticeably lower than this value for VLSFO, and the distribution by ports is fully correlated with the existing emission control zones in the Baltic Sea. Among the quality indicators, only density and viscosity are of research value, since other characteristics (such as sulfur, aluminum and silicon content, flow temperature) are within narrow limits that are included in the ranges required by regulatory documentation. Compared with VLSFO, the viscosity

and density of ULSFO are noticeably lower, which is due to the fact that in order to achieve an ultra-low-sulfur content, residual refined components cannot be used, instead of which hydrotreated vacuum gas oil and low-sulfur diesel fractions with low viscosity and density values are mainly used. Singapore, the largest bunkering port in the world, produces VLSFO low-sulfur marine fuels of various compositions, but the most popular are compositions based on hydrotreated fuel oil (50–70%) and heavy cracker cycle oil (15–35%). Their average content in the released batches is 65% and 18%, respectively.

In the ports of the Middle East, popular VLSFO fuel compositions are based on the available resources of low-sulfur components, namely, hydrotreated fuel oil, the production capacity of which was built not so long ago in this region. The average content of hydrotreated fuel oil in batches is about 87%, which is significantly higher than global values. Additionally, in European ports, due to the specifics of refined oils and the relatively low sulfur content in them, straight-run fuel oil is widely used as components of low-sulfur marine fuels (the average sulfur content is about 0.8%). at 59%. In addition, shipments in Western European ports contain on average significantly more hydrotreated vacuum gas oil (21%) than in the rest of the world (4–5%). Finally, ultra-low-sulfur marine fuels have a very simple basic component composition: hydrotreated vacuum gas oil, hydrotreated diesel fractions, and light catalytic cracker cycle oil. By varying the content of these components, one can obtain all the ULSFO batches recorded in the ports under study. Bunkering with ultra-low-sulfur fuel is carried out overwhelmingly only in Russian and European ports of the Baltic Sea.

**Author Contributions:** Conceptualization, A.E.M. and T.M.M.A.; Methodology, E.S.R.; Validation, D.Y.M.; Formal analysis, V.D.S.; Resources, U.A.M.; Writing—original draft, D.Y.M.; Writing—review & editing, T.M.M.A.; Visualization, V.M.K.; Supervision, V.M.K. and T.M.M.A.; Project administration, M.A.E. All authors have read and agreed to the published version of the manuscript.

**Funding:** This research received no external funding.

**Data Availability Statement:** Not applicable.

**Conflicts of Interest:** The authors declare no conflict of interest.

## Acronyms

| | |
|---|---|
| VLSFO | Very-Low-Sulfur Fuel Oil |
| ULSFO | Ultra-Low-Sulfur Fuel Oil |
| DF | Diesel Fuel |
| VGO | Vacuum Gas Oil |
| HTVGO | Hydrotreated Vacuum Gas Oil |
| LCO | Catalytic Cracker Light Cycle Oil |
| HCO | Catalytic Cracker Heavy Cycle Oil |
| LCGO | Light Coking Gas Oil |
| HCGO | Heavy Coking Gas Oil |
| AR | Atmospheric Residue |
| HDSAR | Hydrodesulfurized Atmospheric Residue |
| FO | Fuel Oil |
| AR | Atmospheric Residue |
| VR | Vacuum Residue |
| IMO | International Maritime Organization |
| SECA | Special Environmental Control Area |
| FAME | Fatty Acid Methyl Ester |
| MFO | Medium Fuel Oil |
| IFO | Intermediate Fuel Oil |
| MDO | Marine Diesel Oil |
| MGO | Marine Gas Oil |

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
