# Peer review of "Technological Potential Analysis and Vacant Technology Forecasting in Properties and Composition of Low-Sulfur Marine Fuel Oil (VLSFO and ULSFO) Bunkered in Key World Ports"

_jmse, doi:10.3390/jmse10121828_

Round 1
Reviewer 1 Report
This manuscript presents a statistical analysis of very-low and ultra-low sulfur fuel oil in different parts of the world. The contents in this manuscript is heavily data-oriented, and a lot of statistical analysis is presented. It looks more like a summary/review of the current low sulfur fuel oils in the world. I suggest that the authors revise before this manuscript can be considered for publication.
Major comments:
1. No keyword is listed in the manuscript. Please revise this part.
2. There are many redundant figures (Fig 10 through 22) that take up a lot of space of the paper. It is recommended that these figures be somehow combined into one, or referred to in the supplementary section.
3. Same goes for some tables. The tables are very long and takes up a lot of space. Consider putting only the key tables in the text and the rest in the supplementary section.
4. The section numbering is weird. There are multiple 3.3.1 sections. Please correct and verify the rest of the sections.
5. It is uncommon that the conclusion is written in a form of bullet points. It is suggested that the conclusion be re-edited to incorporate everything together.
Author Response
Reviewer #1:
This manuscript presents a statistical analysis of very-low and ultra-low sulfur fuel oil in different parts of the world. The contents in this manuscript is heavily data-oriented, and a lot of statistical analysis is presented. It looks more like a summary/review of the current low sulfur fuel oils in the world. I suggest that the authors revise before this manuscript can be considered for publication.
- No keyword is listed in the manuscript. Please revise this part.
- Response: Thank you for your nice suggestion and positive comments which help us to improve the quality of the manuscript, the reviewers’ opinion is highly valuable and appreciated. To answer this question, the keywords have been corrected as the reviewer recommended.
More details in manuscript for these points, please, kindly see in the manuscript:
Keywords: Marine fuel; Very-low sulfur fuel oil (VLSFO); Ultra-low sulfur fuel oil (ULSFO); World ports; Marine energy; Fuel oil
- There are many redundant figures (Fig 10 through 22) that take up a lot of space of the paper. It is recommended that these figures be somehow combined into one, or referred to in the supplementary section.
- Response: I really appreciate your respectful point of view and I would like to thank you for all your efforts and spending time on reviewing our manuscript. To answer this question, the authors want to thank the respected reviewer for great suggestion, but the authors wanted to clarify something that the authors tried to made this before the first submission. The authors found that there was confusion in ideas and vision was not clear to the readers. Therefore, the authors tended to put them in the manuscript in order to make it easier for readers and researchers to understand the subject easily.
- Same goes for some tables. The tables are very long and takes up a lot of space. Consider putting only the key tables in the text and the rest in the supplementary section.
- Response: I really appreciate your respectful point of view and I would like to thank you for all your efforts and spending time on reviewing our manuscript. To answer this question, the authors want to thank the respected reviewer for great suggestion, but the authors wanted to clarify something that the authors tried to made this before the first submission. The authors found that there was confusion in ideas and vision was not clear to the readers. Therefore, the authors tended to put them in the manuscript in order to make it easier for readers and researchers to understand the subject easily.
- The section numbering is weird. There are multiple 3.3.1 sections. Please correct and verify the rest of the sections.
- Response: I really appreciate your respectful point of view and I would like to thank you for all your efforts and spending time on reviewing our manuscript. To answer this question, the authors wanted to introduce their apologize for this mistake, where they are very shamed for this technical mistake. Moreover, they recorrected these mistakes strictly. Thank you extremely much for kind suggestions.
More details in manuscript for these points, please, kindly see in the manuscript:
3.1.1., 3.1.2., 3.1.3., 3.1.4, 3.1.5.
- It is uncommon that the conclusion is written in a form of bullet points. It is suggested that the conclusion be re-edited to incorporate everything together.
- Response: I really appreciate your respectful point of view and I would like to thank you for all your efforts and spending time on reviewing our manuscript. To answer this question, the authors have rewritten the conclusion and incorporate together
Instead of making them in a form of bullet points. Thank you for great recommendation.
More details in manuscript for these points, please, kindly see in the manuscript:
Conclusion section
Reviewer 2 Report
Journal: JMSE (ISSN 2077-1312)
Manuscript ID: jmse-2026865
Title: Technological potential analysis and vacant technology forecasting in properties and composition of low-sulfur marine fuel oil (VLSFO and ULSFO) bunkered in key world ports
Authors: Ershov et al.
In this manuscript, authors have presented a graphical and mathematical analysis for determining properties of the very-low sulfur fuel oil (VLSFO) and ultra-low sulfur fuel oil (ULSFO) in key ports in Asia, Middle East, North America, Western Europe, and Russia is presented. They concluded the following points:
· The key fuel components in Asian ports, the most important of which is Singapore, are hydrodesulfurized atmospheric residues (AR) (50-70%) and catalytic cracker heavy cycle oil (HCO) (15-35%) with the addition of other components, which is explained by the presence of a number of large oil refining centers in the area.
· In the Middle East ports, the most used VLSFO compositions are based on available resources of low-sulfur components, namely hydrodesulfurized AR, the production facilities of which were recently built in the region. The average content of hydrodesulfurized AR in batches is about 87%, which is significantly higher than the average global values.
· In European ports, due to the relatively low sulfur content in processed oils, straight-run AR is widely used as a component of low-sulfur marine fuels. In addition, fuels in Western European ports contain on average significantly more hydrotreated vacuum gas oil (21%) than in the rest of the world (4-5%).
·
In my opinion, the manuscript is original, well-presented, and fit the scope of the JMSE journal. However, there is minor revision, which must be addressed by authors before accepting this paper as follow:
1- Kindly go through whole manuscript and check English language. Correct sentences and phrases where possible.
2- Avoid repeating sentences and educe number of words as possible.
3- If possible, use multi-figure method for combining figures together
Author Response
Reviewer #2:
In this manuscript, authors have presented a graphical and mathematical analysis for determining properties of the very-low sulfur fuel oil (VLSFO) and ultra-low sulfur fuel oil (ULSFO) in key ports in Asia, Middle East, North America, Western Europe, and Russia is presented. They concluded the following points:
- The key fuel components in Asian ports, the most important of which is Singapore, are hydrodesulfurized atmospheric residues (AR) (50-70%) and catalytic cracker heavy cycle oil (HCO) (15-35%) with the addition of other components, which is explained by the presence of a number of large oil refining centers in the area.
- In the Middle East ports, the most used VLSFO compositions are based on available resources of low-sulfur components, namely hydrodesulfurized AR, the production facilities of which were recently built in the region. The average content of hydrodesulfurized AR in batches is about 87%, which is significantly higher than the average global values.
In European ports, due to the relatively low sulfur content in processed oils, straight-run AR is widely used as a component of low-sulfur marine fuels. In addition, fuels in Western European ports contain on average significantly more hydrotreated vacuum gas oil (21%) than in the rest of the world (4-5%).
In my opinion, the manuscript is original, well-presented, and fit the scope of the JMSE journal. However, there is minor revision, which must be addressed by authors before accepting this paper as follow:
- Kindly go through whole manuscript and check English language. Correct sentences and phrases where possible.
- Response: Thank you for your nice suggestion and positive comments which help us to improve the quality of the manuscript, the reviewers’ opinion is highly valuable and appreciated. To answer this question, the manuscript has been revised linguistically by technical academic writer as you recommended.
More details in manuscript for these points, please, kindly see in the manuscript:
Lines: 31,32,33,39,40,41,43,44,45,58,59,60,61,62,63,64,65,66,67,73,76,84,94,95,96,97,106,109,110,114,115,116,117,118,119,120,121,122,123,124,125,130,131,132,133,134,135,136,147,149,150,151,152,153,154,155,156,157,158,159,171,176,181,182,184,193,203,204,205,208,209,210,211,216,217,219,220,225,228,229,233,234,239,243,244,249,252,254,255,256,259,260,265,268,271,273,275,279,281,282,283,285,286,287,288,289,290,291,292,293,294,295,296,297,298,299,300,301,302,303,304,308,309,314,318,320,322,323,324,325,326,330,331,332,333,334,335,336,337,338,339,340,341,342,343,344,346,347,348,350,353,357,358,359,361,365,369,370,372,373,376,385,387,388,389,391,392,394,396,398,399,401,404,406,407,412,413,414,417,421,422,424,426,429,431,435,436,439,442,444,447,449,459,460,463,469,474,476,479,482,485,487,490,496,501,503,504,505,509,510,512,515,518,519,522,524,526,527,528,532,539,540,544,547,548,549,554,555,556,557,558,560,561,,564,565,571,576,581,582,583,587,588,592,593,595,598,600,602,607,608,609,610,611,614,615,619,622,625,628,629,630,635,637,638,639,640,643,644,646,648,649,650,653,653,657,660,669,670,671,672,675,677,685,688,690,700,703,704,705,716,178,720,722,726,727,731,733,735,738,741,742,743,744,752,753,754,758,759,760,769,770,771,775,777,779,781,784,785,795,798,800,801,802,803,804,805,808,813,818,820,82,822,824,828,829,831,833,837,839,840,844,847,849,850,851,854,856,857,861,866,869,871,872,873,877,880,881,883,890,893,895,896,899,900,902,930,931,932,933,934,935,936,937,938,939,940,941,942,943,944,945,946,947,948,949,950.
- Avoid repeating sentences and educe number of words as possible.
- Response: I really appreciate your respectful point of view and I would like to thank you for all your efforts and spending time on reviewing our manuscript. We were really sorry for this careless mistake. To answer this question, the authors have reduced the repeating sentences as possible they can.
- 3. If possible, use multi-figure method for combining figures together
- Response: I really appreciate your respectful point of view and I would like to thank you for all your efforts and spending time on reviewing our manuscript. To answer this question, we understand that the sheer amount of figures in the manuscript is big but exactly because of that it is better not to combine them together since, as it is now, the narration goes smoothly with the graphs and it is easier for the reader to understand the study that way. Moreover, the final layout of the article will be determined by the editors, who will most likely do some of the combining anyway
Reviewer 3 Report
1. Abstract needs to be revised and more data about experimental conclusion needs to be included.
2. In some parts the English needs correction
3. Please include recent papers from last 5 years
4. Title of the Manuscript is too long, try to reduce the title to 10-13 words.
5. In the experimental section, there is lack of real data, FILL IT UP.
6. Make sure all abreviations are properly explained the first time they are mentioned.
7. More detailed description of your method and state-of-art is neede to better understand your work, methodology, and findings.
8. Results need to be rewritten. Add the proper discussion
9. Figures are not clear. replace them with high Quality pictures
Author Response
Reviewer #3: .
- Abstract needs to be revised and more data about experimental conclusion needs to be included.
- Response: Thank you for your nice suggestion and positive comments which help us to improve the quality of the manuscript, the reviewers’ opinion is highly valuable and appreciated. To answer this question, the authors have revised the abstract section, where they deleted some parts and added some parts as the respected reviewer recommended.
More details in manuscript for these points, please, kindly see in the manuscript:
Abstract section
- In some parts the English needs correction
- Response: I really appreciate your respectful point of view and I would like to thank you for all your efforts and spending time on reviewing our manuscript. We were really sorry for this careless mistake. To answer this question, the manuscript has been revised linguistically by technical academic writer as you recommended.
More details in manuscript for these points, please, kindly see in the manuscript:
Line: 31,32,33,39,40,41,43,44,45,58,59,60,61,62,63,64,65,66,67,73,76,84,94,95,96,97,106,109,110,114,115,116,117,118,119,120,121,122,123,124,125,130,131,132,133,134,135,136,147,149,150,151,152,153,154,155,156,157,158,159,171,176,181,182,184,193,203,204,205,208,209,210,211,216,217,219,220,225,228,229,233,234,239,243,244,249,252,254,255,256,259,260,265,268,271,273,275,279,281,282,283,285,286,287,288,289,290,291,292,293,294,295,296,297,298,299,300,301,302,303,304,308,309,314,318,320,322,323,324,325,326,330,331,332,333,334,335,336,337,338,339,340,341,342,343,344,346,347,348,350,353,357,358,359,361,365,369,370,372,373,376,385,387,388,389,391,392,394,396,398,399,401,404,406,407,412,413,414,417,421,422,424,426,429,431,435,436,439,442,444,447,449,459,460,463,469,474,476,479,482,485,487,490,496,501,503,504,505,509,510,512,515,518,519,522,524,526,527,528,532,539,540,544,547,548,549,554,555,556,557,558,560,561,,564,565,571,576,581,582,583,587,588,592,593,595,598,600,602,607,608,609,610,611,614,615,619,622,625,628,629,630,635,637,638,639,640,643,644,646,648,649,650,653,653,657,660,669,670,671,672,675,677,685,688,690,700,703,704,705,716,178,720,722,726,727,731,733,735,738,741,742,743,744,752,753,754,758,759,760,769,770,771,775,777,779,781,784,785,795,798,800,801,802,803,804,805,808,813,818,820,82,822,824,828,829,831,833,837,839,840,844,847,849,850,851,854,856,857,861,866,869,871,872,873,877,880,881,883,890,893,895,896,899,900,902,930,931,932,933,934,935,936,937,938,939,940,941,942,943,944,945,946,947,948,949,950.
- Please include recent papers from last 5 years
- Response: I really appreciate your respectful point of view and I would like to thank you for all your efforts and spending time on reviewing our manuscript. To answer this question, the authors have included recent papers published in the last five years as the respected reviewer recommended.
More details in manuscript for these points, please, kindly see in the manuscript:
References section
- Title of the Manuscript is too long, try to reduce the title to 10-13 words.
- Response: I really appreciate your respectful point of view and I would like to thank you for all your efforts and spending time on reviewing our manuscript. To answer this question, The authors have reduced the title of the manuscript to: (Technological potential analysis in properties and composition of low-sulfur marine fuel oil bunkered in key world ports)
- More details in manuscript for these points, please, kindly see in the manuscript:
Title of manuscript
- 5. In the experimental section, there is lack of real data, FILL IT UP.
- Response: I really appreciate your respectful point of view and I would like to thank you for all your efforts and spending time on reviewing our manuscript. To answer this question, The data about real batches of marine fuels bunkered in the world's largest bunkering ports used for analyzing quality indicators and calculating the most popular VLSFO and ULSFO compositions has been obtained from the Lloyd's Register site. Since this information is private and is given upon paid subscription, we cannot provide the raw information about marine fuel batches. We, however, still provide the processed and analyzed data in the form of graphs, tables, and correlations.
- Make sure all abbreviations are properly explained the first time they are mentioned.
- Response: I really appreciate your respectful point of view and I would like to thank you for all your efforts and spending time on reviewing our manuscript. To answer this question, the authors have checked all abbreviations as the reviewer recommended. Moreover, the authors added some new abbreviation in ACRONYMS section.
More details in manuscript for these points, please, kindly see in the manuscript:
ACRONYMS section
- More detailed description of your method and state-of-art is need to better understand your work, methodology, and findings.
- Response: I really appreciate your respectful point of view and I would like to thank you for all your efforts and spending time on reviewing our manuscript. To answer this question, More detailed description of the methodology used for calculating the most probable fuel compositions is provided in the new paragraph 2.1 – “Methodology”.
More details in manuscript for these points, please, kindly see in the manuscript:
Section 2.1 – “Methodology”.
- 8. Results need to be rewritten. Add the proper discussion
- Response: I really appreciate your respectful point of view and I would like to thank you for all your efforts and spending time on reviewing our manuscript. To answer this question, the authors have already discussed the results all section separately in the all manuscript.
- More details
manuscript
- Figures are not clear. replace them with high Quality pictures
- Response: I really appreciate your respectful point of view and I would like to thank you for all your efforts and spending time on reviewing our manuscript. To answer this question, most of the figures are provided in the form of graphs from MS Excel, meaning that they are actually editable and not just the pictures. This means that the figures are as high quality as they can possibly be. Moreover, the authors have presented an Excel file with most of the graphs, as well as PDF versions of the rest of the figures. So, the authors maximized the Figures as soon as possible.
More details in manuscript for these points, please, kindly see in the manuscript:
All Figures
Reviewer 4 Report
This paper presents a lot of analyzed data on the properties and composition of marine fuels. The production of marine fuels is an important task, because the bulk of the world's cargo transportation is carried out by ships.
1. In the introduction, it is necessary to specify information about the relevance of the research topic. Describe why the authors chose this topic. A little more connection between the introduction and the main part of the article should be added.
2. In the conclusions to the paragraphs or in the conclusion, I would like to see a relationship between the potential composition of the fuel and the impact of this composition on the cost of the fuel, if possible.
3. 20 kg/m3 is recommended to be corrected to kg/m3 on pages 4, 14 (table 2), 20.
4. The abbreviation "ТКГГ" must be translated in some figures (for example, figure 22).
5. In paragraph 3.1. should not be clauses 3.3.1., but should be 3.1.1, 3.1.2, 3.1.3. etc. Similarly, instead of paragraphs 3.3.1 there should be 3.3.1, 3.3.2, 3.3.3. etc.
It is necessary to check for typos in the numbering.
Dear Authors,
I have carefully read the manuscript submitted by you and believe that it fully corresponds to the subject of the Special Issue.
Best regards
Author Response
Reviewer #4:
This paper presents a lot of analyzed data on the properties and composition of marine fuels. The production of marine fuels is an important task, because the bulk of the world's cargo transportation is carried out by ships.
- In the introduction, it is necessary to specify information about the relevance of the research topic. Describe why the authors chose this topic. A little more connection between the introduction and the main part of the article should be added.
- Response: Thank you for your nice suggestion and positive comments which help us to improve the quality of the manuscript, the reviewers’ opinion is highly valuable and appreciated. To answer this question, the authors have added some parts, which mentioned previously by the respected reviewer in the introduction section as the reviewer recommended.
More details in manuscript for these points, please, kindly see in the manuscript:
Introduction section
- In the conclusions to the paragraphs or in the conclusion, I would like to see a relationship between the potential composition of the fuel and the impact of this composition on the cost of the fuel, if possible.
- Response: I really appreciate your respectful point of view and I would like to thank you for all your efforts and spending time on reviewing our manuscript. To answer this question, while analyzing the possible composition of the marine fuels bunkered in different world ports, the principle of the minimal cost was taken into account, meaning that when choosing between two similar fuel components the one that costs less was prioritized. However, calculating the absolute value of the resulting marine fuels is a task that requires additional research. This kind of study will take a lot of time and deserves a publication of its own. The best we can provide is the following cost estimation: AR < HDSAR < HCO < HTVGO < ULSD/DF. So, considering that AR is likely not to be used as it is, due to its high sulphur content, the hydrotreated/hydrodesulphurized AR is supposed to be the cheapest fuel component. However, this assumption only applies to the regions where HDSAR is produced, and in the region where it is not, marine fuel typically does not contain much of it.
More details in manuscript for these points, please, kindly see in the manuscript:
Conclusion section
- 20 kg/m3 is recommended to be corrected to kg/m3 on pages 4, 14 (table 2), 20.
- Response: I really appreciate your respectful point of view and I would like to thank you for all your efforts and spending time on reviewing our manuscript. To answer this question, the authors did not find what the respected reviewer needed on page 4, 14, and 20.
More details in manuscript for these points, please, kindly see in the manuscript:
References section
4.The abbreviation "ТКГГ" must be translated in some figures (for example, figure 22).
- Response: I really appreciate your respectful point of view and I would like to thank you for all your efforts and spending time on reviewing our manuscript. To answer this question, The text and figures in manuscript have been revised so there are no more missed translation points in them.
- 5. In paragraph 3.1. should not be clauses 3.3.1., but should be 3.1.1, 3.1.2, 3.1.3. etc. Similarly, instead of paragraphs 3.3.1 there should be 3.3.1, 3.3.2, 3.3.3. etc.
It is necessary to check for typos in the numbering.
- Response: I really appreciate your respectful point of view and I would like to thank you for all your efforts and spending time on reviewing our manuscript. To answer this question, the authors wanted to introduce their apologize for this mistake, where they are very shamed for this mistake. Moreover, they recorrected these mistakes strictly. Thank you extremely much for kind suggestions.
More details in manuscript for these points, please, kindly see in the manuscript:
3.1.1., 3.1.2., 3.1.3., 3.1.4, 3.1.5.
Round 2
Reviewer 1 Report
I would like to thank the authors for reading the comments and making revisions to the manuscript. After the explanation, I understand the reasons why authors split the graphs and tables. In addition, the conclusion has been modified to integrate the findings in words instead of bullet points.
After careful reviewing, I think this revised manuscript is a more scientifically sound paper than the original version and is ready for publication.
Reviewer 3 Report
No further Comments